# A Game-Theoretic Approach for CSR Emergency Medical Supply Chain during COVID-19 Crisis

**Kefan Xie, Shufan Zhu \* and Ping Gui**

School of Management, Wuhan University of Technology, Wuhan 430070, China; xkf@whut.edu.cn (K.X.); guiping518@whut.edu.cn (P.G.)
\* Correspondence: zhu_shufan@163.com

**Abstract:** The COVID-19 pandemic has caused high fluctuations in the demand for medical supplies. Therefore, emergency medical supplies enterprises have faced challenges in decision making and need to consider more corporate social responsibility (CSR) in production. At the same time, the government needs to take considerable measures to support emergency medical supplies enterprises. As such, our paper researches the decision and coordination problems for emergency medical supply chain considering CSR with the government, manufacturer, and retailer. The manufacturer produces emergency medical supplies. It has additional production technological innovation efforts to improve supply efficiency and assume CSR. The retailer faces uncertain demands and is responsible for undertaking CSR to meet the demands. The government must implement a certain degree of subsidies to ease the impact of the pandemic on emergency medical supply chain enterprises. Meanwhile, our paper further explores the obligations of the economy, society, and efficiency of enterprises under the COVID-19 pandemic and the decision making of enterprises for the implementation of CSR. Based on the principle of maximizing social welfare, we discuss decentralized decision making (without government and with government) and centralized decision making, respectively. On this basis, our paper not only designs a wholesale price–cost sharing joint contract coordination mechanism but also proves that a joint contract can achieve supply chain coordination under certain conditions. Through the analysis, we observe: (1) Government subsidies can improve the enthusiasm of supply chain members to undertake CSR; (2) With the improvement of the retailer's CSR level, the profits of supply chain members and overall performance have improved to a certain degree; (3) To improve supply efficiency and assume social responsibility, the manufacturer implements technological innovation investment. However, it will impose some burden on the manufacturer. Government subsidies allow the manufacturer to balance between social responsibility and its profit.

**Keywords:** emergency medical supply chain coordination; COVID-19; social welfare; wholesale price and cost-sharing joint contract; corporate social responsibility

## 1. Introduction

The global outbreak of the COVID-19 pandemic at the end of 2019 has caused a huge impact on the world economy and increased the risk of global economic recession [1]. According to the United Nations Conference on Trade and Development, the world has lost at least US $3 trillion due to the pandemic and the economic growth rate has dropped by 1.5pp. As of 11 March 2020, there were 118,000 cases in 114 countries around the globe, and 4291 people had died [2]. The spread of the pandemic has disrupted the global supply chain industry and is bound to affect the supply chain [3]. The information uncertainty of the virus, the uniqueness of emergency medical supplies, multiple supply chain sources, pandemics, travel restrictions, and other factors have made emergency management work face higher requirements [4]. The basis of pandemic mitigation and control is the scientific configuration and strong assurance of emergency supplies such as medical protective equipment, medicines, and medical equipment. During the pandemic period, emergency

supplies in pandemic countries have been frequently rushed [5]. For example, Rowan and Laffey (2020) take the republic of Ireland as an example and elaborate on the shortage of personal protective equipment (PPE) arising from the COVID-19 pandemic [6]. Shokrani et al., (2020) indicate the shortage problems of medical and personal protective equipment such as face shields in western countries [7]. In particular, the emergency demand for medical prevention and control supplies causes greater critical power to the supply chain industry, resulting in market mutations in the emergency medical supplies market and significant demand fluctuations [8]. This makes the supply chain unbalanced. However, existing emergency materials are mainly crisis relief equipment, and public health materials production bases are relatively short. To effectively control the pandemic, travel restrictions are implemented, so that product demand and production capacity are not completely matched, and the product delivery rate cannot be guaranteed or maintained [9]. As such, it is imperative to make decisions on supply chain coordination of emergency medical supplies, namely, masks and medical equipment under many uncertain conditions, so that the emergency medical supplies supply chain can be restored from a state of imbalance to a state of coordination.

Due to the imposed quarantine policy and travel restrictions, emergency medical material supply chain companies are encountering the dual pressure of falling incomes and rising operating costs. On the one hand, with the sluggish production of many enterprises and the poor connection between upstream and downstream of the industrial chain, the business volume of enterprises has fallen dramatically [10]. On the other hand, companies are facing rising operating costs in terms of manpower, storage, and pandemic prevention. The essential stores such as masks and disinfectants have also increased costs. At the same time, to stop the spread of the virus, various regions have implemented travel restrictions and set up inspection and disinfection stations, which reduces the efficiency of logistics circulation, increases the transit time, and greatly increases logistic costs. The pandemic has caused large-scale changes in the demand for emergency medical supplies and major changes in production costs, supplies, and incomes. Under the context of multiple pressures, the cash flow of supply chain cooperative companies is becoming challenging and it is difficult to maintain the initially coordinated supply chain. This directly leads to the trend of emergency medical supply chain members reducing production and distribution activities to achieve risk aversion. The government is the ultimate carrier of residual risks in the process of dealing with major public health issues [11]. By building a supply chain coordination platform, the government strengthens coordination efforts and assumes social responsibilities [12]. For users, the government upholds the price of important materials for pandemic prevention and control within an acceptable range; for enterprises, the government has done the following: (1) adopted measures such as reducing or deferring rent and property management fees; (2) subsidized for resumption of work and production and tax and fee reductions to decrease the burden on enterprises; (3) coordinated all member entities to actively assume social responsibilities and maintain sustainable development. For the emergency medical material supply chain, uncertain demand and material supply problems caused by the pandemic make it more challenging for emergency medical members to make decisions. This leads to the tendency of emergency medical member enterprises to lessen production and distribution activities. Nevertheless, to win the sniper battle against the pandemic, emergency medical member enterprises need to consider social responsibilities and ensure materials' supply without losing their profits [13]. For example, manufacturers can alleviate the concerns of downstream members and fulfill their social responsibilities by granting downstream supply chain members to purchase on credit and promising unconditional replacement and no-agreement production. In this way, supply chain members can jointly assume CSR, ensure the supply of emergency medical supplies, and achieve more social welfare. Therefore, based on the uncertainties of the emergency medical supplies supply chain in the epidemic, the coordination of the emergency medical supply chain in terms of corporate social responsibility and government subsidies has become a hot research issue.

In the context of the pandemic, the government, enterprises, and all sectors of society need to be responsible for ensuring the supply of materials as the overall principle to jointly resist the impact of the pandemic. As the main body of handling pandemic incidents, the government can safeguard the output efficiency of the emergency medical asset industry chain through certain subsidy policies to promote the synergy of emergency medical member enterprises, solve the material supply problem, reduce the burden on enterprises, and realize an effective balance between market demand and supply. However, the implementation of the government subsidy policy will not necessarily accomplish optimal decision making and the best profit of emergency medical supplies enterprises. Facing the pandemic, companies are not only members of the emergency medical supplies supply chain but also social participants. The companies have close ties with other stakeholders. Corporate social responsibility (CSR) is a corporate behavior in which an enterprise can realize the social value to other stakeholders, so it can realize its benefits and social value by implementing CSR. For companies, there is a sudden fluctuation in demand, and it is unreasonable to make accurate predictions. However, enterprises will incur certain costs in fulfilling their social responsibilities, and government subsidy policies are an important means of motivating enterprises to fulfill their social responsibilities. Enterprises can also implement CSR to improve the efficiency of government subsidies. There are two ways to consider CSR in supply chain coordination: the CSR level is considered as the investment level of the enterprise or the concern for consumer surplus [14]. Therefore, our paper introduces the social responsibility and the government subsidy and describes the manufacturer's social responsibility level as the corporate social responsibility input level [15]. Then the government subsidizes the company's input and simultaneously describes the retailer's social responsibility level as consumer surplus concerns and analyzes the impact of corporate social responsibility investment and government subsidies on the emergency medical supplies' coordination.

Decisions such as pricing, production planning, and production cost input of any party will influence each other among supply chain members. Therefore, the supply chain coordination mechanism mainly refines decision participants' efficiency. Among the relevant works of literature on supply chain coordination, numerous kinds of literature apply various contracts such as the wholesale price contract [16], revenue sharing contract [17,18], quantity flexibility contract [19,20], option contract [21], buy-back contract [22], and the two-part tariffs contract [23,24] to achieve equilibrium results through supply chain coordination. These types of literature emphasize a single performance measure. In fact, with the continuous evolution of pandemics, a single contract model can no longer coordinate the emergency medical supply chain in complex situations. Therefore, it is impossible to build a supply chain coordination model that only reflects a single goal. To the author's best knowledge, little literature has considered utilizing multi-contract coordination in the emergency medical supply chain when the manufacturer has the effort to develop and improve product quality and the retailer has an implementation level of social responsibility. In addition to considering cost and profit and other destinations or conditions such as service level and efficiency, we can integrate social welfare and utility in all processes as aims of supply chain coordination. When an enterprise decides to take on corporate social responsibility, it considers its profit as well as the best profit of the entire supply chain and social welfare.

More specifically, our research adopts Shu et al. [14] and Li et al. [15]'s work on supply chain coordination from the perspective of the government and enterprises assuming social responsibility. They argue that CSR and government subsidy have positive effects on supply chain decisions and the goal of maximizing social welfare can help increase the profits of supply chain companies. Thus, integrating government subsidy and enterprises' CSR awareness into the emergency medical supply chain and considering their impacts on operational decisions of the emergency medical supply chain are theoretically and empirically meaningful. Most emergency supply chain coordination considers economic benefits, whereas social welfare has rarely been quantitatively examined. To fill this gap,

we model an emergency medical supply chain engaged in CSR and government subsidy composed of one manufacturer and one retailer. The manufacturer fulfills CSR through technological innovation efforts. The retailer fulfills CSR through consumer surplus and may share part of the manufacturer's technological innovation effort cost. Meanwhile, the government improves the determination of manufacturers in technological innovation to fulfill CSR through cost subsidy to the manufacturer. As a result, four emergency medical models including decentralized decision models without and with government subsidies, centralized decision model, and wholesale price–cost sharing joint contract decision models are established. In the following sections, we explore the impacts of CSR implementation level and technological innovation effort on the utilities of emergency medical member enterprises and systems and analyze the relationships among governments, enterprises, and society, to provide insights for government and corporate decision making and have emergency management capabilities against emergencies in the pandemic. In particular, some points are proposed and answered.

1. What are the effects of the retailer's implementation level of social responsibility on profits of emergency medical member enterprises and systems?
2. Will the retailer's implementation level of social responsibility affect government subsidy and consumer surplus?
3. What are the effects of the difficulty factor of manufacturer's technological innovation $\varepsilon$ on profits of emergency medical member enterprises and systems?
4. Can the decentralized supply chain be coordinated and how is it coordinated?

Our paper is structured as follows. Section 2 is a literature analysis. Section 3 makes known the description and fundamental assumptions. Section 4 initially analyses the decentralized decision of emergency medical supplies supply chain without government subsidies and with government subsidies and then builds a game model of centralized decision and adopts a wholesale price–cost sharing joint contract for coordination. Section 5 shows the calculation examples and parameter sensitivity analysis. Section 6 presents the conclusion and brief discussions of future research directions.

## 2. Literature Review

In general, our study summarizes four linked research aspects, namely, emergency supply chain management, social welfare and corporate social responsibility, government intervention, and wholesale price–cost sharing joint contract.

With regard to emergency supply chain management, our paper reviews issues from the outlook of supply chain structure classification, including dual-channel supply chain, reverse supply chain, closed-loop supply chain, and other supply chains. As a good example, on account of the popularity of online sales, a novel product-allocation policy of dual-channel demand under random yield was analyzed. To verify the effectiveness of allocation policy on the supply chain members, the paper investigated the optimal decisions and effects of targets for fulfilling the demand [25]. With the uncertain demand caused by the e-commerce development, the paper derived the optimal pricing, ordering, expected profits of the dual-channel supply chain, and investigated the impacts of relative parameters on supply chain members for the BO and DS policies. The numerical results showed the DS policy was decent for the e-retailers with little market power. If the profit ratio was low, it would be good for the manufacturer [26]. From the perspective of the reverse supply chain, Hong et al., examined the reverse supply chain coordination of electronic products under incomplete information and used game theory to find an equilibrium solution [27]. Mondal and Kumar Roy utilized hybrid facilities to save cost and reduce pollution to cope with uncertain network design about closed-loop supply chain by considering TP and PDRP during the COVID-19 pandemic situation. Then, the paper constructed a stochastic robust correspondence model with chance constraints and reports sensitivity analysis about return rate [28]. In addition, other scholars are concerned with supply chain management. In these areas, because of the destructive effects of the COVID-19 pandemic in the green SC with random demand and limited production capacity, Dehghan-Bonari et al., investigated

SC network consisting of a retailer and two suppliers and utilized the call option contract to enhance the total profit and give more flexibility to the green supplier [29]. Zou, Zou, and Hu examined the low-carbon supply chain consisting of a supplier and a capital-constrained manufacturer with uncertain yield and explored the optimal equilibrium decision. The theoretical results indicated that the supply chain enterprises could choose the trade credit to reach the carbon-emission reduction target [30]. Under symmetric information and asymmetric information scenarios, Jiang, He, and Huang compared the optimal decision of the government penalty in a bioenergy supply chain [31]. Being responsive to demand uncertainty and changes, the study designed a multi-objective economic objective and environmental objective model to minimize total cost, greenhouse gas emissions, water consumption, wastewater disposal, and employee welfare [32]. RezaHoseini, Noori, and Farid Ghannadpour studied the supplier selection problem in the construction supply chain and used the two-objective programming model to evaluate the supply chain. Meanwhile, they set the goals to minimize logistical costs, pollution rate, and greenhouse gas emissions and exploited the Epsilon constraint method to settle up the model. The results showed that proper supplier selection and vehicle selection could contribute to reducing cost, pollution, and project time and promote rescheduling of the project activities [33].

The discussions above show that the previous works on uncertainty in the supply chain include dual-channel supply chain, reverse supply chain, closed-loop supply chain, and other supply chains. Those are the result of emergency incidents. Scholars have seldom considered decision making in the emergency medical supply chain.

Despite much literature about supply chain coordination considering its economic effect, few papers deal with its social aspects. Therefore, our paper integrates the social welfare maximization (SWM) perspective [34,35], consumer surplus [36], CSR, and other social factors into supply chain coordination. For instance, Zhou et al., investigated whether pricing decisions of supply chain members and social welfare were related to carbon tax policies. The paper indicated that social welfare could be effectively boosted by carbon tax regulation under the optimal tax rate [37]. CSR was incorporated into a retailer-dominated sustainability supply chain with the investment of increasing energy efficiency. These results showed that the centralized model could undertake more CSR activities and increase more investment in the technologies to achieve carbon reduction [38]. Su, Weng, and Yang developed a two-stage production system to evaluate the impact of two common corporate social responsibility activities involving social donation and green industry development. The effects analysis derived some insights that spontaneous demand, price-induced demand, and social donation-induced demand were significant features to define whether the CSR activities were effective [39]. Wang et al., analyzed the impacts of the government subsidy, the CSR ratio, and fairness concerns on CLSC operation. The research showed that the CSR was only valid when it was above some threshold. Meanwhile, both CSR and fairness concerns could be beneficial to consumer surplus, the retailer's profit, system profit, and system operating efficiency [40]. Liu et al., explored the CSR investment contribution to the CLSC and considered three modes consisting of: without CSR investment, only one retailer with CSR investment, and both of the competing retailers with CSR investment. Comparing these three models, the research showed that the model of two retailers with CSR investment was the most appropriate for increasing the overall effect [41]. Chen and Ding pointed out that the corporate social responsibilities of enterprises were variable over time because of dynamic and stable cooperative supply chain members and analyzed the impacts of CSR, reputation, and competition in profits. Then simulation analysis indicated that consumer preference for CSR, CSR efficiency, and competition intensity was positively related to CSR efforts [42]. Mondal and C. Giri set the market demand to be affected by CSR investments and developed the C-model and three decentralized models viz. MR-Model, MT-Model, and RT-Model. The study revealed that CSR investments could make MT-Model and MR-Model behave differently and lead to the sustainable development of CLSC [43]. Khosroshahi, Dimitrov, and Reza Hejazi considered the impact of CSR behavior

and the analysis demonstrated that manufacturer CSR decisions could affect the greening degree and transparency level [44].

Reviewing the above literature, we can find out that the CSR of the enterprise is reflected in the improvement of technical level and emission capacity. However, few scholars consider an enterprise's effort to improve product quality into market demand function. Meanwhile, a growing number of studies in the emergency medical supply chain integrate social welfare maximization (SWM), considering CSR, which can be the future research orientation. Meanwhile, the paper explores the impacts of CSR on the optimal strategies, enterprise utilities, and social welfare.

As CSR investments help enterprises reduce burden, some scholars believe that government may help the supply chain accomplish coordination and make decisions. Government intervention methods generally include taxation and subsidies. Some literature considers a single government intervention method. In this area, to reduce greenhouse gas emissions and increase public awareness of environmental issues, the supplier increased the investment in green efforts under government subsidies and the fuel manufacturer confirmed fuel price with the government supervision. Meanwhile, the government supported the production of fuel-efficient cars through the implementation of two policies: (1) Subsidy Tax and (2) Customer Loans. Based on the government intervention, profits of all supply chain members had increased [45]. In addition, Meng et al., considered the effects of government on the dual-channel green supply chain. Numerical comparison illustrated that the government contributed to improving demand and decreasing the sale price for the green products. The government contributed to improving the wholesale price and gradually reducing the sale price and demand for the common product [46]. Rezayat, Yaghoubi, and Fander studied the influence of government intervention in the competitive electronic closed-loop supply chain to support internal industry and presented a new government intervention in which the government attempted to lower price to benefit the end consumer, stop foreign goods imported into the country, and promote domestic prosperity production [47]. Feng, Shen, Zhi, and Pei compared OEM remanufacturing with IR remanufacturing under government subsidies. Numerical analysis indicated that the remanufacturing subsidy could increase remanufacturing capabilities of enterprises and the quality of the new products and remanufacturing products when the remanufacturing was performed by the OEM. However, when the remanufacturing is accomplished by IR, the remanufacturing subsidy is worthless [48]. In addition, some literature considers both subsidies and taxes. Regarding this case, to reduce the environmental burden, the government imposed tax and subsidies into the dynamic CLSC and set different policies on the firm side or consumer side. Then Wu comparatively explored the impacts of different government policies on the profitability of supply chain members and social welfare [49]. The government imposed an environmental tax for producing new and remanufactured products and provided subsidies for remanufacturing products in the CLSC. The numerical analysis showed that when the government fund policy parameters were appropriate, the intervention could be good for the environment, consumer, and society and increase the profits of the CLSC members [50].

It is evident that with a growing body of literature on emergency supply chains, some scholars consider the uncertainty of demand or supply, while others integrate government intervention and consumer surplus. This paper points out that the government subsidizes the cost of enterprise efforts.

Some reviews on the cost-sharing contract to attain supply chain coordination are shown as follows. Liu et al., noted the retailer shared cost ratio of the supplier input effort performance to attain the Pareto optimal and the sustainable development of SC [51]. Zhou et al., presented an emission reduction cost-sharing contract that could attain channel coordination and reach win-win results under certain conditions [52]. He et al., set out to find that the two-way cost-sharing contract could improve the entire SC and member enterprises could select a suitable contract to make the most of expected utilities subject to marginal profits, CSR, and service cost efficiency as well as consumer's low-carbon

preference [53]. Fan et al., studied the impacts of ULQ liability cost-sharing in terms of the product quality, the pricing decision, and the profitability for member enterprises and the entire supply chain in equilibrium [54]. Bai et al., projected revenue and promotional cost-sharing contract (RPS) and a two-part tariff contract (TPT) to attain the supply chain coordination and the coordination effect of TPT was more significant than that of RPS [55]. In addition, Xie et al., combined the revenue-sharing contract with the cost-sharing contract to investigate the CLSC coordination mechanism. The analysis results showed that joint contracts could grow the profits of member enterprises in both the online and offline channels by appropriately sharing ratios and boosting the retailer's effort concerning servicing and recycling [56].

Table 1 summarizes the previous literature related to this paper. To sum up, the above literature has conducted an in-depth analysis of emergency supply chain coordination, social welfare, CSR, and contract coordination and achieved absolute results. However, the above literature has never simultaneously considered government policies, social welfare, CSR, emergency medical supply chain management, and contract coordination. This paper fills the gap by comprehensively integrating these factors into the coordination model of the emergency medical supply chain. Besides, most of the literature analyses emergency supply chain coordination issues, only considering economic and environmental responsibilities. Due to the impact of the pandemic, emergency medical supply chain coordination not only considers economic factors but also focuses on corporate social responsibilities and social welfare. Moreover, our paper shows that the joint contract combining wholesale price with cost sharing can attain the coordination of emergency medical material supply chain considering corporate social responsibility. Therefore, based on government subsidies, a three-stage game model of the manufacturer and the retailer implementing CSR and government participation in decision making is appropriately established, and the impact of the CSR effort of member enterprises on demand, the supply chain itself, and the entire social welfare are comprehensively considered. Besides, due to assuming the CSR, multi-contract is utilized in the emergency medical supply chain with social welfare maximization. Therefore, we can comprehend the effect of government subsidy under the three responsibilities of the economy, CSR and social welfare, and enterprises' decisions to implement social responsibilities from the overall perspective.

**Table 1.** Comparison between this study and prior research.

| Reference | SC Structure | CSR | Demand Influence Factor | Game Approach | Coordination Mechanism | Governments Policies | Social Welfare |
|---|---|---|---|---|---|---|---|
| Chen and Su (2019) | 1M + 1PA | | Price | MS-led PA-led | RS contract | √ | √ |
| Zhou et al., (2018) | 1M + nR | | Price, carbon emissions, and substitutability degree | M-led | | √ | √ |
| Bai et al., (2021) | 1M + 1R | √ | Price, CSR level, and emission technology level | R-led | RCS contract | | |
| Wang et al., (2021) | 1M + 1R | √ | Price | M-led | GSCS contract | √ | |
| Liu et al., (2021) | 1M + 1R | √ | Price, CSR investment level, and competition coefficient between two retailers | M-led | RCS contract | | |

**Table 1.** *Cont.*

| Reference | SC Structure | CSR | Demand Influence Factor | Game Approach | Coordination Mechanism | Governments Policies | Social Welfare |
|---|---|---|---|---|---|---|---|
| Cheng et al., (2021) | 1M + 1R | √ | Price, corporate reputation | M-led | | | |
| Shu et al., (2018) | 1M + 1R | √ | Price | M-led | | | √ |
| Asl-Najaf et al., (2021) | 1M + 1R | | Price and product amount | | TT contract | | |
| Hong et al., (2016) | 1M + 1Re | | | M-led | | | |
| Mondal et al., (2021) | | | Random, COVID-19 | | | | |
| Jiang et al., (2021) | 1PP + nF | | Random | | WP and QP contract | √ | |
| Reza Rezayat et al., (2021) | 2M + 2R | | Price and quality | M-led | | | √ |
| Feng et al., (2021) | OEM + IR | | Random | | | √ | |
| Current study | 1M + 1R | √ | Price, CSR investment level | M-led | WPCS contract | √ | √ |

Note: √ = covered; S = supplier; M = manufacturer; R = retailer; Re = recycler; PA = photovoltaic system assembler; MS = module supplier; PP = power plant; F = farmer; OEM = original equipment manufacturer; IR = independent remanufacturer; RS = revenue sharing; RCS = revenue and cost sharing; GSCS = government subsidy and cost sharing; TT = Two-part tariff; WP = wholesale price; QP = quantity payment; WPCS = wholesale price and cost sharing.

## 3. Model Descriptions and Assumptions

Our paper establishes an emergency medical supplies supply chain system consisting of a manufacturer and a retailer considering the maximization of profits and social welfare. Sudden increase and decrease in the demand for emergency medical supplies due to the pandemic require the manufacturer and the retailer to perform social responsibilities to increase the production of emergency medical supplies. The manufacturer produces emergency medical supplies and has more production technological innovation effort to improve supply efficiency and assume CSR; the retailer faces uncertain demand and is responsible for undertaking CSR to meet demand; to encourage supply chain members to invest in product improvement to enhance supply efficiency and assume CSR, the government also gives manufacturers certain special financial support.

The dynamic game sequence is: (1) The government intends to maximize social welfare and gives the manufacturer certain special subsidies; (2) The dominant manufacturer provides emergency medical supplies, determines the R&D investment of improved products, and determines its wholesale price $P_m$; (3) The retailer in the subordinate position purchases emergency medical supplies from the manufacturer, determines the implementation level of corporate social responsibility (CSR), and sells them at a certain retailer price $P_r$.

Assumption 1. Referencing the literature [38,41,57]. The demand $Q$ is a function of the retailer price and the manufacturer's investment efforts in the emergency medical supply chain, which can be represented as

$$Q = a - \mu P_r + \lambda e \tag{1}$$

where $a, u, \lambda > 0$ and constant; $e > 0$. $a$ stands for the base market size, $\mu$ refers to the sensitivity coefficient of demand to the sale price of emergency medical supplies, $\lambda$ refers to the sensitivity coefficient of demand to manufacturer's effort to improve product design

and improve supply efficiency during the pandemic, and $e$ refers to the manufacturer's effort to enhance product design to improve the efficiency of emergency medical supplies.

Assumption 2. In addition to basic production costs, the manufacturer is working hard on research and development, adopting alternative designs to increase delivery speed to fulfill CSR. Generally speaking, the research and development cost for improved products is nonlinearly increasing over $e$. Assume that the basic production cost is $C_m$ and the additional R&D investment cost is denoted as $\frac{1}{2}\varepsilon e^2$, which is a quadratic cost function. $\varepsilon$ represents the difficulty coefficient of technological innovation. The greater the $\varepsilon$, the more difficult it is to develop alternative designs, and the greater the R&D investment required. This cost function is similar to that in literature [41,58,59]. As the regulator of emergency handling, the government's intervention can affect the supply chain members' decisions and it sets a proportion to share the member's effort cost [15,60,61]. Let $\varphi_m$ represent the government subsidy rate and the government directly subsidizes the manufacturer according to its investment effort, denoted as $\frac{1}{2}\varphi_m \varepsilon e^2$.

Assumption 3. Learning from the implementation of CSR in the references [62,63]. This section discusses how retailers implement CSR and utilize consumer surplus to express the CSR effect. At this point, the retailer aims to maximize its utility $U_r$, $U_r = \pi_r + \beta CS$, where $\pi_r$ is the retailer's profit and $\beta$ is the retailer's concern degree for consumer surplus, that is, is the level of CSR implementation. In addition, consumer surplus refers to the difference between the highest price consumers are willing to pay for the product and the actual market price paid, which is also commonly used in the literature [14,40,64]. Therefore, consumer surplus ($CS$) is:

$$CS = \int_{P_r min}^{P_r max} Q dp = \int_{\frac{a+\lambda e - Q}{\mu}}^{\frac{a+\lambda e}{\mu}} (a - \mu P_r + \lambda e) dp = \frac{Q^2}{2\mu} \tag{2}$$

Social welfare is part of the objective function of the supply chain. To effectively boost the manufacturer to resume work and production, the government intends to maximize social welfare $SW$ and subsidizes effort cost to develop and improve the product quality of the manufacturer. Similar to the literature [37,65], concerning economic assumptions, social welfare consists of four parts: manufacturer's profit $\pi_m$, retailer's profit $\pi_r$, consumer surplus $CS$, and total government subsidy expenditure $GS$. Therefore, the social welfare function is expressed: $SW = \pi_m + \pi_r + CS - GS$, that is

$$\begin{aligned} SW &= (P_r - C_m)Q - \frac{1}{2}(1 - \varphi_m)\varepsilon e^2 + \frac{Q^2}{2\mu} - \frac{1}{2}\varphi_m \varepsilon e^2 \\ &= (P_r - C_m)Q + \frac{Q^2}{2\mu} - \frac{1}{2}\varepsilon e^2 \end{aligned} \tag{3}$$

Besides, $\pi_{sc}$ and $U_{sc}$ represent the profit and utility of the supply chain system, respectively.

## 4. Decision Model of Emergency Medical Supply Chain during the Pandemic

In this section, based on the Stackelberg game, we explore the optimal decisions under the decentralized scenario without or with government subsidy and centralized scenario. For convenience, the superscript $N$, $D$, $C$ represent the case without government subsidy under the decentralized scenario, the case with government subsidy under the decentralized scenario, and the case with government subsidy under the centralized scenario, respectively. Then, based on wholesale price–cost sharing joint contract and government subsidy, we will construct a coordination mechanism considering CSR and government subsidy in the next section.

### 4.1. Manufacturer-Led Game Model in Decentralized Decision Scenario without Government Subsidies

Under decentralized decision without government subsidies, the game order in the emergency medical supplies supply chain is: the manufacturer first sets the wholesale price of emergency medical supplies $P_m$ and the degree of product research and development

efforts, and then the retailer sets the sale price of emergency medical supplies $P_r$. Then the utilities of the manufacturer and the retailer are respectively:

$$U_m{}^N = \pi_m{}^N = (P_m - C_m)(a - \mu P_r + \lambda e) - \frac{1}{2}\varepsilon e^2 \tag{4}$$

$$U_r{}^N = \pi_m{}^N + \beta CS = (P_r - P_m)(a - \mu P_r + \lambda e) + \beta\frac{Q^2}{2\mu} \tag{5}$$

$$SW^N = \pi_m{}^N + \pi_m{}^N + CS - GS = (P_r - C_m)Q - \frac{1}{2}\varepsilon e^2 + \frac{Q^2}{2\mu} \tag{6}$$

According to the reverse derivation method, the optimal wholesale price, the optimal effort to develop and improve product quality, and the optimal sale price under no government subsidies can be obtained as follows:

$$e^{N*} = \frac{\lambda(a - \mu C_m)}{2\mu(2-\beta)\varepsilon - \lambda^2} \tag{7}$$

$$P_m{}^{N*} = \frac{(a + \mu C_m)(2-\beta)\varepsilon - C_m\lambda^2}{[2\mu(2-\beta)\varepsilon - \lambda^2]} \tag{8}$$

$$P_r{}^{N*} = \frac{(3a - 2a\beta + \mu C_m)\varepsilon - C_m\lambda^2}{[2\mu(2-\beta)\varepsilon - \lambda^2]} \tag{9}$$

$$Q^{N*} = \frac{\mu(a - \mu C_m)\varepsilon}{[2\mu(2-\beta)\varepsilon - \lambda^2]} \tag{10}$$

Substituting Equations (7)–(10) into Equations (4)–(6) respectively, $(a - \mu P_r + \lambda e) = \frac{\mu\varepsilon(a-\mu C_m)}{4\mu\varepsilon - (\lambda+\mu\theta)^2}$, we can attain the optimal values under the decentralized decision model without government subsidy:

$$U_m{}^{N*} = \pi_m{}^{N*} = \frac{\varepsilon(a - \mu C_m)^2}{2[2\mu(2-\beta)\varepsilon - \lambda^2]} \tag{11}$$

$$\pi_r{}^{N*} = \frac{(a - \mu C_m)^2(1-\beta)\mu\varepsilon^2}{[2\mu(2-\beta)\varepsilon - \lambda^2]^2} \tag{12}$$

$$CS^{N*} = \frac{\mu(a - \mu C_m)^2\varepsilon^2}{2[2\mu(2-\beta)\varepsilon - \lambda^2]^2} \tag{13}$$

$$U_r{}^{N*} = \pi_r{}^{N*} + \beta CS = \frac{(a - \mu C_m)^2(2-\beta)\mu\varepsilon^2}{2[2\mu(2-\beta)\varepsilon - \lambda^2]^2} \tag{14}$$

$$\pi_{sc}{}^{N*} = \pi_m{}^{N*} + \pi_r{}^{N*} = \frac{\varepsilon(a - \mu C_m)^2[2\mu(3-2\beta)\varepsilon - \lambda^2]}{2[2\mu(2-\beta)\varepsilon - \lambda^2]^2} \tag{15}$$

$$U_{sc}{}^{N*} = \pi_m{}^{N*} + U_r{}^{N*} = \frac{(a - \mu C_m)^2\varepsilon[3(2-\beta)\mu\varepsilon - \lambda^2]}{2[2\mu(2-\beta)\varepsilon - \lambda^2]^2} \tag{16}$$

$$SW^{N*} = \frac{(a - \mu C_m)^2\varepsilon[(7-4\beta)\mu\varepsilon - \lambda^2]}{2[2\mu(2-\beta)\varepsilon - \lambda^2]^2} \tag{17}$$

To ensure members' participation in the emergency medical supply chain, $2\mu(2-\beta)\varepsilon - \lambda^2$ must be greater than 0, otherwise the manufacturer's profit $\pi_m{}^{N*}$ will be less than 0, which will cause the manufacturer to be unwilling to participate in supply chain coordination.

### 4.2. Manufacturer-Led Game Model in Decentralized Decision Scenario with Government Subsidies

To encourage the manufacturer to improve the design of emergency medical supplies to alleviate the shortage of supplies, the government subsidizes the manufacturer's R&D efforts. Therefore, the utilities of emergency medical member enterprises and social welfare can be expressed as:

$$U_m{}^D = \pi_m{}^D = (P_m - C_m)(a - \mu P_r + \lambda e) - \frac{1}{2}(1 - \varphi_m)\varepsilon e^2 \tag{18}$$

$$U_r{}^D = \pi_r{}^D + \beta CS = (P_r - P_m)(a - \mu P_r + \lambda e) + \beta \frac{Q^2}{2\mu} \tag{19}$$

$$SW^D = (P_r - C_m)Q - \frac{1}{2}(1 - \varphi_m)\varepsilon e^2 + \frac{Q^2}{2\mu} - \frac{1}{2}\varphi_m \varepsilon e^2 = (P_r - C_m)Q + \frac{Q^2}{2\mu} - \frac{1}{2}\varepsilon e^2 \tag{20}$$

Taking the reverse derivation method to calculate, we can find the first-order partial derivative of $P_r$ with respect to Equation (19), then:

$$P_r{}^D = \frac{(1 - \beta)(a + \lambda e) + \mu P_m}{(2 - \beta)\mu} \tag{21}$$

where $(a - \mu P_r + \lambda e) = \frac{(a + \lambda e) - \mu P_m}{(2 - \beta)}$.

Substituting Equation (21) into Equation (18), we can get $U_m = (P_m - C_m)\frac{(a + \lambda e) - \mu P_m}{(2 - \beta)} - \frac{1}{2}(1 - \varphi_m)\varepsilon e^2$.

Calculating the Hessian matrix of $U_m$ concerning $P_m$ and $e$, we can obtain

$$H^D = \begin{bmatrix} \frac{\partial^2 U_m}{\partial^2 P_m} & \frac{\partial^2 U_m}{\partial P_m e} \\ \frac{\partial^2 U_m}{\partial e P_m} & \frac{\partial^2 U_m}{\partial^2 e} \end{bmatrix} = \begin{bmatrix} -\frac{2\mu}{(2 - \beta)} & \frac{\lambda}{(2 - \beta)} \\ \frac{\lambda}{(2 - \beta)} & -(1 - \varphi_m)\varepsilon \end{bmatrix} \tag{22}$$

where $\left| H^D \right| = \frac{2\mu(1 - \varphi_m)\varepsilon(2 - \beta) - \lambda^2}{(2 - \beta)^2} > 0$ and $-\frac{2\mu}{(2 - \beta)} < 0$, the Hessian matrix is negative definite, then $U_m$ is a stringently joint concave function about $P_m$ and $e$. Calculating the first-order partial derivatives of $U_m$ with respect to $P_m$ and $e$, we can express the optimal wholesale price and the optimal effort to develop and improve product quality in the decentralized decision scenario.

$$P_m{}^D = \frac{(a + \mu C_m)(2 - \beta)(1 - \varphi_m)\varepsilon - C_m \lambda^2}{[2\mu(2 - \beta)(1 - \varphi_m)\varepsilon - \lambda^2]} \tag{23}$$

$$e^D = \frac{\lambda a - \mu \lambda C_m}{2\mu(2 - \beta)(1 - \varphi_m)\varepsilon - \lambda^2} \tag{24}$$

Substituting Equations (23) and (24) into Equation (21), we can get

$$P_r{}^D = \frac{(3a - 2a\beta + \mu C_m)(1 - \varphi_m)\varepsilon - C_m \lambda^2}{[2\mu(2 - \beta)(1 - \varphi_m)\varepsilon - \lambda^2]} \tag{25}$$

Substituting $P_r{}^{D*}$ and $e^{D*}$ into Equation (20) to find the first-order partial derivative of $\varphi_m$, we can get the government subsidy ratio in the decentralized decision scenario:

$$\varphi_m{}^{D*} = \frac{3 - 2\beta}{7 - 4\beta} \tag{26}$$

Then the optimal wholesale price, the optimal effort to develop and improve product quality, and the optimal sale price of decentralized decision scenario are expressed as:

$$P_m^{D*} = \frac{2\varepsilon(a + \mu C_m)(2 - \beta)^2 - C_m\lambda^2(7 - 4\beta)}{4\mu\varepsilon(2 - \beta)^2 - (7 - 4\beta)\lambda^2} \tag{27}$$

$$e^{D*} = \frac{\lambda(a - \mu C_m)(7 - 4\beta)}{4\mu\varepsilon(2 - \beta)^2 - (7 - 4\beta)\lambda^2} \tag{28}$$

$$P_r^{D*} = \frac{2\varepsilon(3a - 2a\beta + \mu C_m)(2 - \beta) - C_m\lambda^2(7 - 4\beta)}{4\mu\varepsilon(2 - \beta)^2 - (7 - 4\beta)\lambda^2} \tag{29}$$

$$Q^{D*} = \frac{2\mu\varepsilon(a - \mu C_m)(2 - \beta)}{4\mu\varepsilon(2 - \beta)^2 - (7 - 4\beta)\lambda^2} \tag{30}$$

Substituting Equations (27)–(30) into Equations (18)–(20) respectively, $(a - \mu P_r + \lambda e) = \frac{\mu\varepsilon(a - \mu C_m)(2 - 4\beta)}{4\mu\varepsilon(2 - \beta)^2 - (7 - 4\beta)\lambda^2}$, we can get the following optimal decisions with government subsidy.

$$U_m^{D*} = \pi_m^{D*} = \frac{\varepsilon(a - \mu C_m)^2(2 - \beta)}{[4\mu\varepsilon(2 - \beta)^2 - (7 - 4\beta)\lambda^2]} \tag{31}$$

$$\pi_r^{D*} = \frac{4\mu\varepsilon^2(a - \mu C_m)^2(2 - \beta)^2(1 - \beta)}{[4\mu\varepsilon(2 - \beta)^2 - (7 - 4\beta)\lambda^2]^2} \tag{32}$$

$$CS^{D*} = \frac{2\mu\varepsilon^2(a - \mu C_m)^2(2 - \beta)^2}{[4\mu\varepsilon(2 - \beta)^2 - (7 - 4\beta)\lambda^2]^2} \tag{33}$$

$$U_r^{D*} = \pi_r^{D*} + \beta CS = \frac{2\mu\varepsilon^2(a - \mu C_m)^2(2 - \beta)^3}{[4\mu\varepsilon(2 - \beta)^2 - (7 - 4\beta)\lambda^2]^2} \tag{34}$$

$$\pi_{sc}^{D*} = \pi_m^{D*} + \pi_r^{D*} = \frac{\varepsilon(a - \mu C_m)^2(2 - \beta)[4\mu\varepsilon(2 - \beta)(3 - 2\beta) - (7 - 4\beta)\lambda^2]}{[4\mu\varepsilon(2 - \beta)^2 - (7 - 4\beta)\lambda^2]^2} \tag{35}$$

$$U_{sc}^{D*} = \pi_m^{D*} + U_r^{D*} = \frac{(a - \mu C_m)^2(2 - \beta)\varepsilon[6\mu\varepsilon(2 - \beta)^2 - (7 - 4\beta)\lambda^2]}{[4\mu\varepsilon(2 - \beta)^2 - (7 - 4\beta)\lambda^2]^2} \tag{36}$$

$$SW^{D*} = \frac{(a - \mu C_m)^2(7 - 4\beta)\varepsilon}{2[4\mu\varepsilon(2 - \beta)^2 - (7 - 4\beta)\lambda^2]} \tag{37}$$

$$GS^{D*} = \frac{1}{2}\varphi_m\varepsilon e^2 = \frac{\lambda^2\varepsilon(a - \mu C_m)^2(7 - 4\beta)(3 - 2\beta)}{2[4\mu\varepsilon(2 - \beta)^2 - (7 - 4\beta)\lambda^2]^2} \tag{38}$$

### 4.3. Manufacturer-Led Centralized Decision Game Model

Based on the centralized decision scenario, the government first determines the subsidy coefficient $\varphi_m$ for the manufacturer according to the maximization of social welfare, and then the emergency medical material supply chain decision maker determines the effort to develop and improve product quality $e$ and the sale price $P_r$. Therefore, the overall utility function of the emergency medical supply chain $U_{sc}^c$ and social welfare $SW^c$ functions are expressed as:

$$U_{sc}^c = \pi_{sc}^c = (P_r - C_m)(a - \mu P_r + \lambda e) - \frac{1}{2}(1 - \varphi_m)\varepsilon e^2 \tag{39}$$

$$SW^c = (P_r - C_m)Q + \frac{Q^2}{2\mu} - \frac{1}{2}\varepsilon e^2 \tag{40}$$

Calculating the Hessian matrix of $U_{sc}{}^c$ concerning $P_r$ and $e$, we can obtain

$$H^c = \begin{bmatrix} \frac{\partial^2 U_{sc}}{\partial^2 P_r} & \frac{\partial^2 U_{sc}}{\partial P_r e} \\ \frac{\partial^2 U_{sc}}{\partial e P_r} & \frac{\partial^2 U_{sc}}{\partial^2 e} \end{bmatrix} = \begin{bmatrix} (\beta - 2)\mu & \lambda(1 - \beta) \\ \lambda(1 - \beta) & -(1 - \varphi_m)\varepsilon + \frac{\beta}{\mu}\lambda\lambda \end{bmatrix} \tag{41}$$

where $|H^c| = (2 - \beta)\mu(1 - \varphi_m)\varepsilon - \lambda^2 > 0$ and $(\beta - 2)\mu < 0$, the Hessian matrix is negative definite, then $U_{sc}{}^c$ is a stringently joint concave function about $P_r$ and $e$. Calculating the first-order partial derivatives of $U_{sc}{}^c$ with respect to $P_r$ and $e$, we can express the optimal sale price and the optimal effort to develop and improve product quality in the centralized decision model.

$$P_r = \frac{[a(1 - \beta) + \mu C_m](1 - \varphi_m)\varepsilon - C_m \lambda^2}{[(2 - \beta)(1 - \varphi_m)\varepsilon\mu - \lambda^2]} \tag{42}$$

$$e = \frac{(a - \mu C_m)\lambda}{[(2 - \beta)(1 - \varphi_m)\varepsilon\mu - \lambda^2]} \tag{43}$$

Substituting $P_r{}^*$ and $e^*$ into Equation (40) to find the first-order partial derivative of $\varphi_m$, we can get the optimal government subsidy ratio under the centralized decision model:

$$\varphi_m{}^{c*} = \frac{(1 - \beta)}{(3 - 2\beta)} \tag{44}$$

The optimal decisions of emergency supply chain system under centralized decision making are:

$$P_r{}^{c*} = \frac{(2 - \beta)[a(1 - \beta) + \mu C_m]\varepsilon - (3 - 2\beta)C_m \lambda^2}{(2 - \beta)^2 \varepsilon\mu - (3 - 2\beta)\lambda^2} \tag{45}$$

$$e^{c*} = \frac{(3 - 2\beta)(a - \mu C_m)\lambda}{(2 - \beta)^2 \varepsilon\mu - (3 - 2\beta)\lambda^2} \tag{46}$$

$$Q^{c*} = \frac{(2 - \beta)\mu(a - \mu C_m)\varepsilon}{(2 - \beta)^2 \varepsilon\mu - (3 - 2\beta)\lambda^2} \tag{47}$$

$$\pi_{sc}{}^{c*} = \frac{\varepsilon(2 - \beta)(a - \mu C_m)^2 [2(1 - \beta)(2 - \beta)\mu\varepsilon - \lambda^2(3 - 2\beta)]}{2[(2 - \beta)^2 \varepsilon\mu - (3 - 2\beta)\lambda^2]^2} \tag{48}$$

$$CS^{c*} = \frac{(2 - \beta)^2 \mu(a - \mu C_m)^2 \varepsilon^2}{2[(2 - \beta)^2 \varepsilon\mu - (3 - 2\beta)\lambda^2]^2} \tag{49}$$

$$U_{sc}{}^{c*} = \frac{\varepsilon(2 - \beta)(a - \mu C_m)^2}{2[(2 - \beta)^2 \varepsilon\mu - (3 - 2\beta)\lambda^2]} \tag{50}$$

$$SW^{c*} = \frac{(3 - 2\beta)(a - \mu C_m)^2 \varepsilon}{2[(2 - \beta)^2 \varepsilon\mu - (3 - 2\beta)\lambda^2]} \tag{51}$$

$$GS^{c*} = \frac{(3 - 2\beta)(1 - \beta)\varepsilon(a - \mu C_m)^2 \lambda^2}{2[(2 - \beta)^2 \varepsilon\mu - (3 - 2\beta)\lambda^2]^2} \tag{52}$$

Summarizing the optimal decisions under the above three scenarios, we can obtain Table 2.

**Table 2.** The optimal decision under three scenarios.

| Variable | Without Government Subsidy | Decentralized Decision | Centralized Decision |
|---|---|---|---|
| $P_m$ | $\frac{(a+\mu C_m)(2-\beta)\varepsilon - C_m\lambda^2}{[2\mu(2-\beta)\varepsilon - \lambda^2]}$ | $\frac{2\varepsilon(a+\mu C_m)(2-\beta)^2 - C_m\lambda^2(7-4\beta)}{4\mu\varepsilon(2-\beta)^2 - (7-4\beta)\lambda^2}$ | |
| $P_r$ | $\frac{(3a-2a\beta+\mu C_m)\varepsilon - C_m\lambda^2}{[2\mu(2-\beta)\varepsilon - \lambda^2]}$ | $\frac{2\varepsilon(3a-2a\beta+\mu C_m)(2-\beta) - C_m\lambda^2(7-4\beta)}{4\mu\varepsilon(2-\beta)^2 - (7-4\beta)\lambda^2}$ | $\frac{(2-\beta)[a(1-\beta)+\mu C_m]\varepsilon - (3-2\beta)C_m\lambda^2}{(2-\beta)^2\varepsilon\mu - (3-2\beta)\lambda^2}$ |
| $e$ | $\frac{\lambda(a-\mu C_m)}{2\mu(2-\beta)\varepsilon - \lambda^2}$ | $\frac{\lambda(a-\mu C_m)(7-4\beta)}{4\mu\varepsilon(2-\beta)^2 - (7-4\beta)\lambda^2}$ | $\frac{(3-2\beta)(a-\mu C_m)\lambda}{(2-\beta)^2\varepsilon\mu - (3-2\beta)\lambda^2}$ |
| $Q$ | $\frac{\mu(a-\mu C_m)\varepsilon}{[2\mu(2-\beta)\varepsilon - \lambda^2]}$ | $\frac{2\mu\varepsilon(a-\mu C_m)(2-\beta)}{4\mu\varepsilon(2-\beta)^2 - (7-4\beta)\lambda^2}$ | $\frac{(2-\beta)\mu(a-\mu C_m)\varepsilon}{(2-\beta)^2\varepsilon\mu - (3-2\beta)\lambda^2}$ |
| $\varphi m$ | | $\frac{3-2\beta}{7-4\beta}$ | $\frac{(1-\beta)}{(3-2\beta)}$ |
| $\pi_r$ | $\frac{(a-\mu C_m)^2(1-\beta)\mu\varepsilon^2}{[2\mu(2-\beta)\varepsilon - \lambda^2]^2}$ | $\frac{4\varepsilon\mu\varepsilon(a-\mu C_m)^2(2-\beta)^2(1-\beta)}{[4\mu\varepsilon(2-\beta)^2 - (7-4\beta)\lambda^2]^2}$ | |
| $U_r$ | $\frac{(a-\mu C_m)^2(2-\beta)\mu\varepsilon^2}{2[2\mu(2-\beta)\varepsilon - \lambda^2]^2}$ | $\frac{2\mu\varepsilon^2(a-\mu C_m)^2(2-\beta)^3}{[4\mu\varepsilon(2-\beta)^2 - (7-4\beta)\lambda^2]^2}$ | |
| $\pi_m$ | $\frac{\varepsilon(a-\mu C_m)^2}{2[2\mu(2-\beta)\varepsilon - \lambda^2]}$ | $\frac{\varepsilon(a-\mu C_m)^2(2-\beta)}{[4\mu\varepsilon(2-\beta)^2 - (7-4\beta)\lambda^2]}$ | |
| $\pi_{sc}$ | $\frac{\varepsilon(a-\mu C_m)^2[2(3-2\beta)\mu\varepsilon - \lambda^2]}{2[2\mu(2-\beta)\varepsilon - \lambda^2]^2}$ | $\frac{\varepsilon(a-\mu C_m)^2(2-\beta)[4\mu\varepsilon(2-\beta)(3-2\beta) - (7-4\beta)\lambda^2]}{[4\mu\varepsilon(2-\beta)^2 - (7-4\beta)\lambda^2]^2}$ | $\frac{\varepsilon(2-\beta)(a-\mu C_m)^2[2(1-\beta)(2-\beta)\mu\varepsilon - \lambda^2(3-2\beta)]}{2[(2-\beta)^2\varepsilon\mu - (3-2\beta)\lambda^2]^2}$ |
| $CS$ | $\frac{\mu(a-\mu C_m)^2\varepsilon^2}{2[2\mu(2-\beta)\varepsilon - \lambda^2]^2}$ | $\frac{2\varepsilon\mu\varepsilon(a-\mu C_m)^2(2-\beta)^2}{[4\mu\varepsilon(2-\beta)^2 - (7-4\beta)\lambda^2]^2}$ | $\frac{(2-\beta)^2\mu(a-\mu C_m)^2\varepsilon^2}{2[(2-\beta)^2\varepsilon\mu - (3-2\beta)\lambda^2]^2}$ |
| $U_{sc}$ | $\frac{(a-\mu C_m)^2\varepsilon[3(2-\beta)\mu\varepsilon - \lambda^2]}{2[2\mu(2-\beta)\varepsilon - \lambda^2]^2}$ | $\frac{(a-\mu C_m)^2(2-\beta)\varepsilon[6\mu\varepsilon(2-\beta)^2 - (7-4\beta)\lambda^2]}{[4\mu\varepsilon(2-\beta)^2 - (7-4\beta)\lambda^2]^2}$ | $\frac{\varepsilon(2-\beta)(a-\mu C_m)^2}{2[(2-\beta)^2\varepsilon\mu - (3-2\beta)\lambda^2]}$ |

**Corollary 1.** (1) $\frac{\partial P_m^{D*}}{\partial\lambda} > 0$; $\frac{\partial\varphi_m^{D*}}{\partial\lambda} = 0$; $\frac{\partial P_r^{D*}}{\partial\lambda} > 0$; $\frac{\partial e^{D*}}{\partial\lambda} < 0$; $\frac{\partial Q^{D*}}{\partial\lambda} > 0$; (2) $\frac{\partial\pi_m^{D*}}{\partial\lambda} > 0$; $\frac{\partial\pi_r^{D*}}{\partial\lambda} > 0$; $\frac{\partial\pi_{sc}^{D*}}{\partial\lambda} > 0$; $\frac{\partial CS^{D*}}{\partial\lambda} > 0$; $\frac{\partial SW^{D*}}{\partial\lambda} > 0$.

Corollary 1 shows that under decentralized decision making, the government's subsidy coefficient to the manufacturer has no direct relationship with the sensitivity of demand to the manufacturers' efforts to improve product design and supply efficiency ($\lambda$), but $\lambda$ has stimulated manufacturer's motivation for research and development. The manufacturer has accelerated research and development so that product cost has increased and the wholesale price has increased. At the same time, market demand has also increased, and the dual effects of price and market demand have increased the profits of emergency medical member enterprises, and social welfare has also increased.

**Corollary 2.** *(1) The wholesale price $P_m^{D*}$, the effort to develop and improve product quality $e^{D*}$, and the retailer's order quantity $Q^{D*}$ are negatively correlated with the manufacturer's technical innovation difficulty coefficient $\varepsilon$, that is, $\frac{\partial P_m^{D*}}{\partial\varepsilon} < 0$; $\frac{\partial e^{D*}}{\partial\varepsilon} < 0$; $\frac{\partial Q^{D*}}{\partial\varepsilon} < 0$;*

*(2) The manufacturer's optimal profit, retailer's optimal utility, consumer surplus, and total social welfare are negatively correlated the manufacturer's technological innovation difficulty coefficient $\varepsilon$, namely $\frac{\partial\pi_m^{D*}}{\partial\varepsilon} < 0$; $\frac{\partial U_r^{D*}}{\partial\varepsilon} < 0$; $\frac{\partial CS^{D*}}{\partial\varepsilon} < 0$; $\frac{\partial SW^{D*}}{\partial\varepsilon} < 0$; when $\varepsilon > \frac{(7-4\beta)\lambda^2}{2\mu(2-\beta)^2}$, the retailer's economic profit is positively correlated with the manufacturer's technological innovation difficulty coefficient $\varepsilon$; when $\frac{(7-4\beta)\lambda^2}{4\mu(2-\beta)^2} < \varepsilon < \frac{(7-4\beta)\lambda^2}{2\mu(2-\beta)^2}$, the retailer's economic profit is negatively correlated with the technological innovation difficulty coefficient $\varepsilon$.*

**Proof of Corollary 2.** Calculating the partial derivative of $P_m^{D*}$, $e^{D*}$, and $Q^{D*}$ with respect to $\varepsilon$, we can obtain:

$$\frac{\partial P_m^{D*}}{\partial\varepsilon} = \frac{-2(2-\beta)^2(a-\mu C_m)(7-4\beta)\lambda^2}{[4\mu\varepsilon(2-\beta)^2 - (7-4\beta)\lambda^2]^2} < 0, \quad \frac{\partial e^{D*}}{\partial\varepsilon} = -\frac{4\mu(2-\beta)^2\lambda(a-\mu C_m)(7-4\beta)}{[4\mu\varepsilon(2-\beta)^2 - (7-4\beta)\lambda^2]^2} < 0,$$

$$\frac{\partial Q^{D*}}{\partial \varepsilon} = \frac{-2\mu(7-4\beta)\lambda^2(a-\mu C_m)(2-\beta)}{\left[4\mu\varepsilon(2-\beta)^2-(7-4\beta)\lambda^2\right]^2} < 0, \quad \frac{\partial \pi_m{}^{D*}}{\partial \varepsilon} = \frac{-(7-4\beta)\lambda^2(a-\mu C_m)^2(2-\beta)}{\left[4\mu\varepsilon(2-\beta)^2-(7-4\beta)\lambda^2\right]^2} < 0,$$

$$\frac{\partial U_r{}^{D*}}{\partial \varepsilon} = \frac{-2(7-4\beta)\lambda^2\mu(a-\mu C_m)^2(2-\beta)^3}{\left[4\mu\varepsilon(2-\beta)^2-(7-4\beta)\lambda^2\right]^3} < 0, \quad \frac{\partial CS^{D*}}{\partial \varepsilon} = \frac{-2(7-4\beta)\lambda^2\mu(a-\mu C_m)^2(2-\beta)^2}{\left[4\mu\varepsilon(2-\beta)^2-(7-4\beta)\lambda^2\right]^3} < 0,$$

$$\frac{\partial SW^{D*}}{\partial \varepsilon} = \frac{-(7-4\beta)^2\lambda^2(a-\mu C_m)^2}{2\left[4\mu\varepsilon(2-\beta)^2-(7-4\beta)\lambda^2\right]^2} < 0, \quad \frac{\partial \pi_r{}^{D*}}{\partial \varepsilon} =$$
$$\frac{8\mu\varepsilon(a-\mu C_m)^2(2-\beta)^2(1-\beta)\left[2\mu\varepsilon(2-\beta)^2-(7-4\beta)\lambda^2\right]}{\left[4\mu\varepsilon(2-\beta)^2-(7-4\beta)\lambda^2\right]^3}$$

For $\frac{\partial \pi_r{}^{D*}}{\partial \varepsilon}$, when $\varepsilon > \frac{(7-4\beta)\lambda^2}{2\mu(2-\beta)^2}$, we can obtain $\frac{\partial \pi_r{}^{D*}}{\partial \varepsilon} > 0$. Namely, when $\varepsilon > \frac{(7-4\beta)\lambda^2}{2\mu(2-\beta)^2}$, the retailer's economic profit is positively correlated with the technological innovation difficulty coefficient $\varepsilon$.

Similarly, when $\frac{(7-4\beta)\lambda^2}{4\mu(2-\beta)^2} < \varepsilon < \frac{(7-4\beta)\lambda^2}{2\mu(2-\beta)^2}$, we can obtain $\frac{\partial \pi_r{}^{D*}}{\partial \varepsilon} < 0$ namely, when $\frac{(7-4\beta)\lambda^2}{4\mu(2-\beta)^2} < \varepsilon < \frac{(7-4\beta)\lambda^2}{2\mu(2-\beta)^2}$, the retailer's economic profit is positively correlated with the technological innovation difficulty coefficient $\varepsilon$. $\square$

Corollary 2 indicates that as the manufacturer's technological innovation difficulty coefficient $\varepsilon$ increases, the overall production cost of products increases, resulting in the decrease of manufacturer's enthusiasm for development and improvement of product quality and a decrease in market demand. Therefore, it is difficult to increase product demand by increasing the manufacturer's effort to develop and improve product quality, but it is possible to stimulate market demand by reducing the manufacturer's wholesale price.

Simultaneously, as the technological innovation difficulty coefficient $\varepsilon$ increases, the cost of the manufacturer's development and improvement of product quality has increased, and the efficiency of the manufacturer's development and improvement of the product has relatively decreased. When the manufacturer's technological innovation difficulty coefficient $\varepsilon$ is in a small interval, the decline in the efficiency of the manufacturer's development and improvement of product reduces the retailer's economic profit. When the manufacturer's technological innovation difficulty coefficient $\varepsilon$ is in a big interval, the retailer can implement CSR to reduce the rate of decline in the manufacturer's efficiency of development and improvement, thereby increasing the retailer's economic profit.

In general, the increase in the manufacturer's technological innovation difficulty coefficient $\varepsilon$ makes the manufacturer's profit and consumer welfare decline more than the potential increase of the retailer's economic profit, so the retailer's overall utility and social welfare have declined.

**Corollary 3.** *(1)* $\frac{\partial P_m{}^{D*}}{\partial \beta} > 0$; $\frac{\partial \varphi_m{}^{D*}}{\partial \beta} < 0$; $\frac{\partial e^{D*}}{\partial \beta} > 0$; *when* $\varepsilon > \frac{(25-28\beta+8\beta^2)\lambda^2}{4\mu(2-\beta)^2}$, $\frac{\partial P_r{}^{D*}}{\partial \beta} < 0$, *when* $\varepsilon \le \frac{(25-28\beta+8\beta^2)\lambda^2}{4\mu(2-\beta)^2}$, $\frac{\partial P_r{}^{D*}}{\partial \beta} \ge 0$; $\frac{\partial Q^{D*}}{\partial \beta} > 0$;

*(2)* $\frac{\partial \pi_m{}^{D*}}{\partial \beta} > 0$; *when* $\varepsilon \le \frac{\lambda^2(12-4\beta^2-13\beta)}{4\mu\beta(2-\beta)^2}$, $\frac{\partial \pi_r{}^{D*}}{\partial \beta} > 0$; *when* $\varepsilon > \frac{\lambda^2(12-4\beta^2-13\beta)}{4\mu\beta(2-\beta)^2}$, $\frac{\partial \pi_r{}^{D*}}{\partial \beta} < 0$; *There is a critical value* $\underline{\beta} \approx 0.95$, *if* $0 \le \beta \le \underline{\beta}$, $\frac{\partial \pi_{sc}{}^{D*}}{\partial \beta} \ge 0$, *if* $\underline{\beta} < \beta \le 1$, *when* $\varepsilon \le \lambda^2 \frac{(4-3\beta)-\sqrt{(2-\beta)[-96+\beta(208-143\beta+32\beta^2)]}}{16\mu(2-\beta)^2(1-\beta)}$ *or* $\varepsilon \ge \lambda^2 \frac{(4-3\beta)+\sqrt{(2-\beta)[-96+\beta(208-143\beta+32\beta^2)]}}{16\mu(2-\beta)^2(1-\beta)}$, $\frac{\partial \pi_{sc}{}^{D*}}{\partial \beta} \ge 0$, *when* $\lambda^2 \frac{(4-3\beta)-\sqrt{(2-\beta)[-96+\beta(208-143\beta+32\beta^2)]}}{16\mu(2-\beta)^2(1-\beta)} < \varepsilon < \lambda^2 \frac{(4-3\beta)+\sqrt{(2-\beta)[-96+\beta(208-143\beta+32\beta^2)]}}{16\mu(2-\beta)^2(1-\beta)}$, $\frac{\partial \pi_{sc}{}^{D*}}{\partial \beta} < 0$; $\frac{\partial CS^{D*}}{\partial \beta} > 0$; $\frac{\partial SW^{D*}}{\partial \beta} > 0$.

**Proof of Corollary 3.** $\frac{\partial P_m{}^{D*}}{\partial \beta} = \frac{4\varepsilon(a-\mu C_m)(2-\beta)\lambda^2(3-2\beta)}{[4\mu\varepsilon(2-\beta)^2-(7-4\beta)\lambda^2][4\mu\varepsilon(2-\beta)^2-(7-4\beta)\lambda^2]} > 0; \ \frac{\partial \varphi_m{}^{D*}}{\partial \beta} = \frac{-2}{(7-4\beta)^2}$
$< 0; \ \frac{\partial e^{D*}}{\partial \beta} = \frac{24\mu\varepsilon(2-\beta)\lambda(a-\mu C_m)}{[4\mu\varepsilon(2-\beta)^2-(7-4\beta)\lambda^2]^2} > 0;$

$\frac{\partial P_r{}^{D*}}{\partial \beta} = \frac{2\varepsilon(a-\mu C_m)[(25-28\beta+8\beta^2)\lambda^2-4\mu\varepsilon(2-\beta)^2]}{[4\mu\varepsilon(2-\beta)^2-(7-4\beta)\lambda^2]^2}$, when $\varepsilon > \frac{(25-28\beta+8\beta^2)\lambda^2}{4\mu(2-\beta)^2}$, $\frac{\partial P_r{}^{D*}}{\partial \beta} < 0$, when

$\varepsilon \le \frac{(25-28\beta+8\beta^2)\lambda^2}{4\mu(2-\beta)^2}$, $\frac{\partial P_r{}^{D*}}{\partial \beta} \ge 0$;

$\frac{\partial \pi_r{}^{D*}}{\partial \beta} = \frac{4(2-\beta)\mu\varepsilon^2(a-\mu C_m)^2[-4\mu\beta(2-\beta)^2-\lambda^2(4\beta^2+13\beta-12)]}{[4\mu\varepsilon(2-\beta)^2-(7-4\beta)\lambda^2]^3}$, when $\varepsilon \le \frac{\lambda^2(12-4\beta^2-13\beta)}{4\mu\beta(2-\beta)^2}$,

$\frac{\partial \pi_r{}^{D*}}{\partial \beta} \ge 0$, when $\varepsilon > \frac{\lambda^2(12-4\beta^2-13\beta)}{4\mu\beta(2-\beta)^2}$, $\frac{\partial \pi_r{}^{D*}}{\partial \beta} < 0$.

Let $\frac{\partial \pi_{sc}{}^{D*}}{\partial \beta} = \frac{\varepsilon(a-\mu C_m)^2[32\mu^2\varepsilon^2(2-\beta)^3(1-\beta)+4\mu\varepsilon(2-\beta)\lambda^2(3\beta-4)+(7-4\beta)\lambda^4]}{[4\mu\varepsilon(2-\beta)^2-(7-4\beta)\lambda^2]^3} > 0$, where $h = 32\mu^2\varepsilon^2(2-\beta)^3(1-\beta) + 4\mu\varepsilon(2-\beta)\lambda^2(3\beta-4) + (7-4\beta)\lambda^4$. Solving the quadratic function of $h$ with respect to $\varepsilon$ and according to the discriminant $\Delta = 16\mu^2(2-\beta)^3\lambda^4[(3\beta-4)^2 - 8(7-4\beta)(2-\beta)(1-\beta)] = 16\mu^2(2-\beta)^3\lambda^4[-96 + \beta(208-143\beta+32\beta^2)]$, we can obtain: (1) when $0 \le \beta \le \underline{\beta} \approx 0.95$, $\Delta \le 0$, $\frac{\partial \pi_{sc}{}^{D*}}{\partial \beta} \ge 0$; (2) when $\underline{\beta} < \beta \le 1$ and $\varepsilon \le \lambda^2 \frac{(4-3\beta)-\sqrt{(2-\beta)[-96+\beta(208-143\beta+32\beta^2)]}}{16\mu(2-\beta)^2(1-\beta)}$ or $\varepsilon \ge \lambda^2 \frac{(4-3\beta)+\sqrt{(2-\beta)[-96+\beta(208-143\beta+32\beta^2)]}}{16\mu(2-\beta)^2(1-\beta)}$, $\frac{\partial \pi_{sc}{}^{D*}}{\partial \beta} \ge 0$; when $\underline{\beta} < \beta \le 1$ and $\lambda^2 \frac{(4-3\beta)-\sqrt{(2-\beta)[-96+\beta(208-143\beta+32\beta^2)]}}{16\mu(2-\beta)^2(1-\beta)} < \varepsilon < \lambda^2 \frac{(4-3\beta)+\sqrt{(2-\beta)[-96+\beta(208-143\beta+32\beta^2)]}}{16\mu(2-\beta)^2(1-\beta)}$, $\frac{\partial \pi_{sc}{}^{D*}}{\partial \beta} < 0$. □

Corollary 3 demonstrates that with the enhancement of social responsibility implementation level $\beta$, (1) the government subsidy coefficient decreases, indicating that the implementation of social responsibility $\beta$ can improve the efficiency of government subsidies. (2) The effort of the manufacturer to develop and improve product quality has increased. On the one hand, the wholesale price of emergency medical supplies has increased, and on the other hand, market demand has increased and consumer surplus has increased. As the retailer implements CSR, the manufacturer's profit has increased. (3) For the retailer, the smaller the difficulty coefficient of technological innovation $\varepsilon$, the easier it is for the manufacturer to develop and improve product quality and the higher the effort degree to develop and improve product quality. At this time, as the retailer's social responsibility implementation level $\beta$ increases, the retailer's sales price increases, and thus profits increase. Conversely, the greater the difficulty degree of technological innovation $\varepsilon$, the higher the manufacturer's cost of developing and improving products, which will affect the degree of effort in developing and improving product quality. Therefore, the sale price decreases, leading to a decrease in the retailer's profit. (4) When the retailer's implementation level of social responsibility β is within a certain range, the overall profit of the supply chain will increase. Otherwise, when β is too high, the overall profit trend depends on the change of manufacturer's technological innovation difficulty coefficient $\varepsilon$.

Combining Corollary 2 and Corollary 3, the retailer's economic profit and overall system profits are affected by the manufacturer's technological innovation difficulty coefficient $\varepsilon$. and the retailer's social responsibility implementation level $\beta$. Therefore, supply chain members should weigh the proportions of R&D and improving product design investment and the degree of CSR implementation when making decisions.

**Corollary 4.** *In the decentralized decision scenario with government subsidies, the optimal effort of the manufacturer to develop and improve product quality, the manufacturer's optimal profit, the retailer's optimal economic profit, and the optimal social welfare are higher than corresponding values in the decentralized decision scenario without government subsidies, namely, $e^{D*} > e^{N*}$; $\pi_m{}^{D*} > \pi_m{}^{N*}$; $\pi_r{}^{D*} > \pi_r{}^{N*}$; $SW^{D*} > SW^{N*}$.*

**Proof of Corollary 4.**

$$\Delta e^* = e^{D*} - e^{N*} = \frac{\lambda(a - \mu C_m)\mu\varepsilon(2 - \beta)(3 - 2\beta)}{[2\mu(2 - \beta)\varepsilon - \lambda^2][4\mu\varepsilon(2 - \beta)^2 - (7 - 4\beta)\lambda^2]} > 0$$

$$\Delta\pi_m^* = \pi_m^{D*} - \pi_m^{N*} = \frac{(5 - 3\beta)\lambda^2\varepsilon(a - \mu C_m)^2}{2[2\mu(2 - \beta)\varepsilon - \lambda^2][4\mu\varepsilon(2 - \beta)^2 - (7 - 4\beta)\lambda^2]} > 0$$

$$\Delta\pi_r^* = \pi_r^{D*} - \pi_r^{N*} = \frac{4(6 - 4\beta)(2 - \beta)\lambda^2\varepsilon\mu\varepsilon(a - \mu C_m)^2(1 - \beta)}{[4\mu\varepsilon(2 - \beta)^2 - (7 - 4\beta)\lambda^2]^2[2\mu\varepsilon(2 - \beta) - \lambda^2]} > 0$$

$$\Delta SW^* = SW^{D*} - SW^{N*} = \frac{\mu\varepsilon^2\lambda^2(2\beta - 3)^2(a - \mu C_m)^2}{2[4\mu\varepsilon(2 - \beta)^2 - (7 - 4\beta)\lambda^2][2\mu(2 - \beta)\varepsilon - \lambda^2]^2} > 0 \ \square$$

Corollary 4 shows that compared with no government subsidy, the manufacturer under government subsidy has a higher effort to improve product design and improve supply efficiency, indicating that the manufacturer can improve product design quality with government subsidy. Simultaneously, government subsidies can promote profits and social welfare. Although the government has not implemented subsidies for the retailer, the subsidy to the manufacturer increases social welfare and indirectly boosts the retailer's profit. This is because when the government provides a subsidy for R&D, it is equivalent to reducing the cost of the manufacturer's improved product, which boosts the manufacturer's profit. After the manufacturer gains more profit, it can encourage the manufacturer to conduct more product development and fulfill social responsibilities. With the increase in market demand, the retailer will further increase its profit margin and will be more active in assuming social responsibilities.

**Corollary 5.** *The optimal effort of the manufacturer to develop and improve product quality, the optimal order quantity, the optimal social welfare, and the optimal overall utility in centralized decision scenario are greater than the corresponding values in decentralized decision scenario, namely, $e^{c*} > e^{D*}, Q^{c*} > Q^{D*}, SW^{c*} > SW^{D*}, U_{sc}^{c*} > U_{sc}^{D*}$.*

**Proof of Corollary 5.**

$$e^{c*} - e^{D*} = \frac{(5 - 4\beta)(a - \mu C_m)\lambda\mu\varepsilon(2 - \beta)^2}{[4\mu\varepsilon(2 - \beta)^2 - (7 - 4\beta)\lambda^2][(2 - \beta)^2\varepsilon\mu - (3 - 2\beta)\lambda^2]} > 0,$$

$$Q^{c*} - Q^{D*} = \frac{2\mu\varepsilon(2 - \beta)^2 - \lambda^2}{[4\mu\varepsilon(2 - \beta)^2 - (7 - 4\beta)\lambda^2][(2 - \beta)^2\varepsilon\mu - (3 - 2\beta)\lambda^2]} > 0,$$

$$SW^{c*} - SW^{D*} = \frac{(5 - 4\beta)(a - \mu C_m)^2\mu\varepsilon^2(2 - \beta)^2}{2[4\mu\varepsilon(2 - \beta)^2 - (7 - 4\beta)\lambda^2][(2 - \beta)^2\varepsilon\mu - (3 - 2\beta)\lambda^2]} > 0,$$

$$U_{sc}^{c*} - U_{sc}^{D*} = \varepsilon(a - \mu C_m)^2(2 - \beta)\frac{4\mu\varepsilon\mu\varepsilon(2 - \beta)^4 - 6\mu\varepsilon(2 - \beta)^2\lambda^2 + (7 - 4\beta)\lambda^4}{2[(2 - \beta)^2\varepsilon\mu - (3 - 2\beta)\lambda^2][4\mu\varepsilon(2 - \beta)^2 - (7 - 4\beta)\lambda^2]^2}.$$

Let $U_{sc}^{c*} - U_{sc}^{D*} = \frac{\varepsilon(a - \mu C_m)^2(2 - \beta)[4\mu^2\varepsilon^2(2 - \beta)^4 - 6\mu\varepsilon(2 - \beta)^2\lambda^2 + (7 - 4\beta)\lambda^4]}{2[(2 - \beta)^2\varepsilon\mu - (3 - 2\beta)\lambda^2][4\mu\varepsilon(2 - \beta)^2 - (7 - 4\beta)\lambda^2]^2} > 0$, where $f = 4\mu\varepsilon\mu\varepsilon(2 - \beta)^4 - 6\mu\varepsilon(2 - \beta)^2\lambda^2 + (7 - 4\beta)\lambda^4$. Solving the quadratic function of $f$ with respect to $\varepsilon$ and according to the discriminant $\Delta = 4\mu^2(2 - \beta)^4\lambda^4(16\beta - 19)$, we can obtain when $0 \leq \beta \leq 1, \Delta < 0, U_{sc}^{c*} > U_{sc}^{D*}$. $\square$

*4.4. Supply Chain Coordination Model under the Wholesale Price–Cost Sharing Joint Contract*

Improved product design requires the manufacturer to invest the corresponding cost, so the retailer can share part of the manufacturer's input cost for improved products. At the same time, for both parties to increase profits, the manufacturer and the retailer need to adjust prices separately. Therefore, the wholesale price–cost sharing contract can realize the coordination of the emergency medical supplies supply chain. Under this joint contract, the manufacturer first provides a benchmark wholesale price $P_{m0}$, and then the retailer shares part of the manufacturer's development and improvement costs for products, with a share ratio of $\omega$. Therefore, the utility function and social welfare are respectively in the wholesale price–cost sharing joint contract scenario:

$$U_m(\omega) = \pi_m(\omega) = (P_{m0} - C_m)(a - \mu P_r + \lambda e) - \frac{1}{2}(1 - \omega)(1 - \varphi_m)\varepsilon e^2 \tag{53}$$

$$U_r(\omega) = \pi_r(\omega) + \beta CS = (P_r - P_{m0})(a - \mu P_r + \lambda e) + \beta \frac{Q^2}{2\mu} - \frac{1}{2}\omega(1 - \varphi_m)\varepsilon e^2 \tag{54}$$

$$SW(\omega) = \pi_m(\omega) + \pi_r(\omega) + CS - GS = (P_r - C_m)Q - \frac{1}{2}\varepsilon e^2 + \frac{Q^2}{2\mu} \tag{55}$$

First, considering the incentive compatibility constraint, the government subsidizes the manufacturer's R&D input cost according to the optimal subsidy coefficient in the centralized decision scenario, that is, $\varphi_m(\omega) = \varphi_m{}^* = \frac{(1-\beta)}{(3-2\beta)}$. The reverse derivation method is adopted, and the optimal sale price and the manufacturer's optimal effort to develop and improve product quality are respectively:

$$P_r(\omega)^* = \frac{(1 - \omega)(2 - \beta)\varepsilon[(3 - 2\beta)a + \mu C_m] - (3 - 2\beta)C_m\lambda^2}{[2(2 - \beta)^2(1 - \omega)\mu\varepsilon - (3 - 2\beta)\lambda^2]} \tag{56}$$

$$e(\omega)^* = \frac{(3 - 2\beta)\lambda(a - \mu C_m)}{[2(2 - \beta)^2(1 - \omega)\mu\varepsilon - (3 - 2\beta)\lambda^2]} \tag{57}$$

$$P_{m0}(\omega)^* = \frac{(2 - \beta)^2(a + \mu C_m)(1 - \omega)\varepsilon - (3 - 2\beta)\lambda^2 C_m}{[2(2 - \beta)^2(1 - \omega)\mu\varepsilon - (3 - 2\beta)\lambda^2]} \tag{58}$$

The optimal decision values in the wholesale price–cost sharing joint contract scenario can be expressed as:

$$U_m(\omega)^* = \pi_m(\omega)^* = \frac{(a - \mu C_m)^2(1 - \omega)(2 - \beta)\varepsilon}{2[2(2 - \beta)^2(1 - \omega)\mu\varepsilon - (3 - 2\beta)\lambda^2]} \tag{59}$$

$$\pi_r(\omega) = (a - \mu C_m)^2(2 - \beta)\varepsilon \frac{2(1 - \omega)^2(2 - \beta)\varepsilon(1 - \beta)\mu - (3 - 2\beta)\lambda^2\omega}{2[2(2 - \beta)\mu(2 - \beta)(1 - \omega)\varepsilon - (3 - 2\beta)\lambda^2]^2} \tag{60}$$

$$U_r(\omega)^* = \pi_r(\omega)^* + \beta CS(\omega)^* = (a - \mu C_m)^2(2 - \beta)\varepsilon \frac{(1 - \omega)^2(2 - \beta)^2\mu\varepsilon - (3 - 2\beta)\lambda^2\omega}{2[2(2 - \beta)^2(1 - \omega)\mu\varepsilon - (3 - 2\beta)\lambda^2]^2} \tag{61}$$

$$CS(\omega)^* = \frac{(a - \mu C_m)^2(2 - \beta)^2\mu\varepsilon^2(1 - \omega)^2}{[2(2 - \beta)^2(1 - \omega)\mu\varepsilon - (3 - 2\beta)\lambda^2]^2} \tag{62}$$

$$SW(\omega)^* = \frac{(a - \mu C_m)^2\varepsilon[3(2 - \beta)^3\varepsilon\mu(1 - \omega)^2 - (3 - 2\beta)^2\lambda^2]}{[2(2 - \beta)\mu(2 - \beta)(1 - \omega)\varepsilon - (3 - 2\beta)\lambda^2]^2} \tag{63}$$

To enable the wholesale price–cost sharing contract to achieve coordination, $\pi_m(\omega)^* > \pi_m^{D*}$ and $\pi_r(\omega)^* > \pi_r^{D*}$. From the two inequalities, we can get

$$\begin{cases} \omega \geq \frac{1}{(7-4\beta)} \\ (1-\omega)^2(2-\beta)^2\mu\varepsilon - (3-2\beta)\lambda^2\omega \geq \frac{4\mu\varepsilon(2-\beta)^2[2(2-\beta)^2\mu\varepsilon(1-\omega)-(3-2\beta)\lambda^2]^2}{[4\mu\varepsilon(2-\beta)^2-(7-4\beta)\lambda^2]^2} \end{cases} \tag{64}$$

Equation (64) indicates that when the cost-sharing coefficient $\omega$ satisfies the above conditions, the wholesale price–cost sharing contract can coordinate the emergency medical supplies supply chain and realize effective distribution of economic benefits among supply chain members.

Finally, we will compare our work with previous studies. Our findings confirm that when the government subsidizes manufacturer's technological innovation effort, the profits of supply chain members and social welfare have increased. When the government's subsidy for the manufacturer's technological innovation effort reach a certain value, the utilities of supply chain members after contract coordination are higher than that of decentralized decision making and reach the level of centralized decision making [11]. We also extend this research by considering CSR. Our research also confirms that appropriate effort of undertaking CSR can increase the profits of supply chain members and social welfare [13]. Besides, our work also confirms that revenue sharing-cost sharing can lead to perfect coordination [55]. However, our work discusses the manufacturer's cost-sharing, not the retailer's cost-sharing. In general, we utilize the game model to extend this research in view of government subsidy, CSR, social welfare, and COVID-19 and enrich emergency medical supply chain operation decisions under the pandemic.

## 5. Numerical Comparison of Different Cases and Parameters Sensitivity Analysis

To further display the above results and verify the validity of the joint contract model, our paper implements sensitivity analysis on some important parameters using numerical simulation. The parameters are summarized in the form of two datasets in Table 3. Because of the limited access to real market data, the first datasets are set in accordance with those in previous studies [41,66], which satisfy the hypothetical conditions in the theoretical model. The second datasets applied in this study have been selected as a control group.

**Table 3.** Values of parameters under two datasets.

| Parameters | $a$ | $\mu$ | $\lambda$ | $C_m$ | $\varepsilon$ |
|---|---|---|---|---|---|
| First example | 100 | 1 | 0.5 | 10 | 1 |
| Second example | 10,000 | 12 | 10 | 200 | 16 |

### 5.1. Numerical Analysis of Equilibrium Results

Substituting related parameters into the decentralized decision scenario, the corresponding optimal results are shown in Tables 4 and 5 under two datasets.

Table 4 displays that the wholesale price $P_m$, the effort to develop and improve product quality $e$, the manufacturer's profit $\pi_m$, consumer surplus $CS$, government subsidies $GS$, and social welfare $SW$ increase with the increase of the CSR implementation level $\beta$ under the decentralized decision scenario. On the contrary, the government subsidy coefficient $\varphi_m$ decreases accordingly under two datasets. Reviewing the first example in Table 3, when $\varepsilon = 1, 0 < \beta < 1$ and $\varepsilon > \frac{(25-28\beta+8\beta^2)\lambda^2}{4\mu(2-\beta)^2}$, the retailer's sale price $P_r$ is inversely proportional to the retailer's social responsibility implementation level $\beta$. When $0 < \beta < 0.33724$ and $\varepsilon = 1 < \frac{\lambda^2(12-4\beta^2-13\beta)}{4\mu\beta(2-\beta)^2}$, the retailer's profit $\pi_r$ is proportional to the social responsibility implementation level $\beta$. When $0.33724 < \beta < 1$ and $\varepsilon = 1 > \frac{\lambda^2(12-4\beta^2-13\beta)}{4\mu\beta(2-\beta)^2}$, the retailer's profit $\pi_r$ is inversely proportional to the retailer's social responsibility implementation

level $\beta$. When $\beta \in [0.1, 0.9]$ and $\frac{\partial \pi_{sc}^{D*}}{\partial \beta} \geq 0$, the total system profit $\pi_{sc}$ is proportional to the social responsibility implementation level $\beta$. Therefore, Corollary 3 is supported by the results of Table 4.

**Table 4.** Impact of retailer's CSR level $\beta$ on equilibrium results.

| | | Decision Variables | | | | Profits and Social Welfare | | | | | | |
|---|---|---|---|---|---|---|---|---|---|---|---|---|
| | $\beta$ | $P_m$ | $P_r$ | $e$ | $\varphi_m$ | $\pi_m$ | $\pi_r$ | $U_r$ | $\pi_{sc}$ | $CS$ | $GS$ | $SW$ |
| First example | 0.1 | 60.81 | 84.87 | 23.22 | 0.424 | 1203.3 | 643.51 | 679.26 | 1846.8 | 357.50 | 228.76 | 163.40 |
| | 0.2 | 61.11 | 83.83 | 24.45 | 0.42 | 1277.8 | 645.07 | 725.71 | 1922.9 | 403.17 | 250.74 | 192.87 |
| | 0.3 | 61.45 | 82.64 | 25.82 | 0.41 | 1362.0 | 641.27 | 778.68 | 2003.3 | 458.05 | 275.78 | 229.82 |
| | 0.4 | 61.83 | 81.27 | 27.33 | 0.407 | 1457.8 | 629.70 | 839.60 | 2087.5 | 524.75 | 276.72 | 304.40 |
| | 0.5 | 62.26 | 79.68 | 29.03 | 0.40 | 1567.7 | 606.87 | 910.30 | 2174.6 | 606.87 | 337.15 | 337.15 |
| | 0.6 | 62.73 | 77.80 | 30.94 | 0.39 | 1695.1 | 567.56 | 993.22 | 2262.6 | 709.45 | 374.63 | 416.26 |
| | 0.7 | 63.28 | 75.57 | 33.10 | 0.38 | 1844.1 | 503.83 | 1091.6 | 2348.0 | 839.71 | 417.37 | 521.71 |
| | 0.8 | 63.89 | 72.87 | 35.55 | 0.37 | 2020.8 | 403.31 | 1210.0 | 2424.1 | 1008.3 | 465.64 | 665.19 |
| | 0.9 | 64.59 | 69.55 | 38.35 | 0.35 | 2233.1 | 246.25 | 1354.4 | 2479.3 | 1231.3 | 518.97 | 864.94 |
| Second example | 0.1 | 615.60 | 812.47 | 237.45 | 0.42 | 831,210 | 516,740 | 545,450 | 1,347,900 | 287,080 | 19,135 | 683.40 |
| | 0.3 | 628.69 | 805.21 | 268.86 | 0.41 | 958,250 | 534,160 | 648,620 | 1,492,400 | 381,540 | 23,929 | 997.04 |
| | 0.5 | 645.60 | 794.14 | 309.44 | 0.40 | 1,128,900 | 529,500 | 794,250 | 1,658,400 | 529,500 | 30,642 | 1532.1 |
| | 0.7 | 668.16 | 776.20 | 363.59 | 0.38 | 1,368,500 | 466,880 | 1,011,600 | 1,835,400 | 778,140 | 40,288 | 2518.0 |
| | 0.9 | 699.38 | 744.77 | 438.50 | 0.35 | 1,725,100 | 247,310 | 1,360,200 | 1,972,400 | 1,236,600 | 54,292 | 4524.3 |

**Table 5.** Impact of manufacturer's technological innovation difficulty coefficient $\varepsilon$ on equilibrium results ($\beta = 0.25$).

| | | Decision Variables | | | | Profits and Social Welfare | | | | | | |
|---|---|---|---|---|---|---|---|---|---|---|---|---|
| | $\varepsilon$ | $P_m$ | $P_r$ | $e$ | $\varphi_m$ | $\pi_m$ | $\pi_r$ | $U_r$ | $\pi_{sc}$ | $CS$ | $GS$ | $SW$ |
| First example | 0.4 | 74.85 | 102.65 | 79.41 | 0.417 | 1667.6 | 1030 | 1201.7 | 2697.7 | 686.69 | 1051.0 | 840.83 |
| | 0.6 | 66.54 | 90.77 | 46.15 | 0.417 | 1453.8 | 782.84 | 913.31 | 2236.7 | 521.89 | 532.54 | 426.04 |
| | 0.8 | 63.13 | 85.90 | 32.53 | 0.417 | 1366.3 | 691.36 | 806.59 | 2057.6 | 460.91 | 352.74 | 282.19 |
| | 1 | 61.28 | 83.26 | 25.12 | 0.417 | 1318.6 | 643.97 | 751.30 | 1962.6 | 429.31 | 262.85 | 210.28 |
| | 1.2 | 60.11 | 81.59 | 20.45 | 0.417 | 1288.6 | 615.03 | 717.53 | 1903.7 | 410.02 | 209.19 | 167.36 |
| | 1.4 | 59.31 | 80.45 | 17.25 | 0.417 | 1268.1 | 595.54 | 694.79 | 1863.6 | 397.03 | 173.63 | 138.90 |
| | 1.6 | 58.73 | 79.61 | 14.92 | 0.417 | 1253.0 | 581.52 | 678.44 | 1834.6 | 387.68 | 148.35 | 118.68 |
| Second example | 12 | 679.90 | 885.57 | 391.75 | 0.417 | 1,042,100 | 676,800 | 789,600 | 1,718,900 | 451,200 | 38,368 | 1534.7 |
| | 14 | 646.98 | 838.55 | 312.76 | 0.417 | 970,590 | 587,150 | 685,000 | 1,557,700 | 391,430 | 28,530 | 1141.2 |
| | 16 | 625.11 | 807.31 | 260.27 | 0.417 | 923,110 | 531,100 | 619,620 | 1,454,200 | 354,070 | 22,581 | 903.23 |
| | 18 | 609.53 | 785.04 | 222.87 | 0.417 | 889,270 | 492,880 | 575,020 | 1,382,100 | 328,590 | 18,627 | 745.09 |
| | 20 | 597.86 | 768.38 | 194.87 | 0.417 | 863,930 | 465,190 | 542,730 | 1,329,100 | 310,130 | 15,823 | 632.92 |

According to the above inferences, the greater the manufacturer's technological innovation difficulty coefficient $\varepsilon$, the more difficult it is to implement the retailer's social responsibility. To comply with the content of this research, it is assumed that the manufacturer's technological innovation difficulty coefficient $\varepsilon$ is relatively small, that is when $0 < \beta < 0.33724$, $\varepsilon \leq \frac{\lambda^2 (12 - 4\beta^2 - 13\beta)}{4\mu\beta(2-\beta)^2}$ holds.

In Table 5, when $\beta = 0.25$, as the manufacturer's technological innovation difficulty coefficient $\varepsilon$ increases, the wholesale price $P_m$, the sale price $P_r$, the effort to develop and improve product quality $e$, and the profit $\pi_m$, the profit $\pi_r$, supply chain system profit $\pi_{sc}$, consumer surplus $CS$, total government subsidy expenditure $GS$, and social welfare $SW$ all decrease accordingly under two datasets. It shows that the technological innovation of the manufacturer to improve products caused by the pandemic directly affects the reduction of the equilibrium value of the emergency medical supplies supply chain. To maintain the balance, the government as the main body of the pandemic can subsidize the technological innovation of the manufacturer. The results of Table 5 are the same as those in Corollary 2. Besides, as long as certain conditions are met, the trends of the two datasets are consistent; subsequent studies take the first datasets as an example.

To further display the effect of the cost-sharing coefficient $\omega$ on equilibrium results, assume that the retailer's social welfare concern $\beta = 0.25$, and from Equation (64) we can see that when $\omega$ satisfies $0.17 \leq \omega \leq 0.45$, the wholesale price–cost sharing contract can achieve the coordination of emergency medical supplies supply chain. In the decentralized decision scenario, the manufacturer's effort to improve the product quality $e$, the optimal profit $\pi_m$, and the optimal utility $U_r$ are 25.12, 1318.6, and 751.30, respectively. It can be seen from Table 6 that when $\omega$ is between $[0.17, 0.45]$, the optimal profit $\pi_m$ and the optimal utility $U_r$ in the joint contract decision scenario are respectively greater than the corresponding values in the decentralized decision scenario. Therefore, the emergency medical member enterprises can accept the joint contract. The results of Table 6 are the same as those in Equation (64).

**Table 6.** Impact of cost-sharing coefficient $\omega$ on equilibrium results.

| $\omega$ | $P_m$ | $P_r$ | $e$ | $\pi_m$ | $\pi_r$ | $U_r$ | $\pi_{sc}$ | $CS$ | $SW$ |
|---|---|---|---|---|---|---|---|---|---|
| 0.17 | 61.31 | 83.30 | 25.23 | 2638.7 | 1213.6 | 1428.5 | 3852.3 | 859.59 | 3876.2 |
| 0.22 | 61.77 | 83.96 | 27.09 | 2662.6 | 1199.8 | 1418.7 | 3862.5 | 875.25 | 3861.1 |
| 0.27 | 62.31 | 84.73 | 29.25 | 2690.3 | 1178.7 | 1402.1 | 3869.0 | 893.58 | 3835.8 |
| 0.32 | 62.94 | 85.64 | 31.78 | 2722.9 | 1146.8 | 1375.6 | 3869.6 | 915.32 | 3795.5 |
| 0.37 | 63.70 | 86.71 | 34.79 | 2761.6 | 1098.8 | 1334.2 | 3860.4 | 941.52 | 3732.7 |
| 0.42 | 64.61 | 88.01 | 38.43 | 2808.4 | 1026.4 | 1269.8 | 3834.7 | 973.70 | 3635.1 |
| 0.45 | 65.25 | 88.93 | 41.00 | 2841.5 | 965.59 | 1214.8 | 3807.0 | 996.78 | 3551.9 |

In addition, to compare results in two cases of no government subsidies and government subsidies, we substitute related parameters into two decision situations of decentralized decisions: no government subsidies and government subsidies and obtain the optimum consequences as shown in Table 7 in two cases of no government subsidies and government subsidies.

Comparing the values in Table 7 in two cases, we can get the following analysis: In comparison with non-government subsidies, when the government subsidizes cooperative companies, the wholesale price, manufacturer's effort to develop and improve product quality, the respective utility, and overall utility utilities of emergency medical member enterprises, consumer surplus, and social welfare have improved, which indicates that the government can inspire supply chain cooperative companies to actively assume social responsibilities and improve the efficiency of material supply through subsidies, thus influencing the utility of supply chain cooperative companies.

**Table 7.** The consequences in two cases of no government subsidies and government subsidies ($\beta = 0.25$).

|  | Without Government Subsidies | With Government Subsidies |
|:---:|:---:|:---:|
| **Variables** | **Decentralized Scenario** | **Decentralized Scenario** |
| $P_m$ | 58.46 | 61.28 |
| $P_r$ | 79.23 | 83.26 |
| $e$ | 13.85 | 25.12 |
| $U_m$ | 1246.2 | 1318.6 |
| $U_r$ | 671.01 | 751.30 |
| $U_{sc}$ | 1917.2 | 2069.9 |
| $CS$ | 383.43 | 429.31 |
| $SW$ | 2204.7 | 2260.5 |

*5.2. Impact of Retailer's CSR Level on Equilibrium Results*

To further show the impact of the implementation level of social responsibility $\beta$ on the equilibrium results, we can, assuming $\beta$ fluctuates between 0 and 0.2, discuss the impact of the implementation level of social responsibility $\beta$ on other parameters under the three types of decentralized decision without government subsidies, the decentralized decision with government subsidies, and joint contract. Figure 1 illustrates the impact of the implementation level of social responsibility $\beta$ on the manufacturer's effort to develop and improve product quality, the wholesale price $P_m$, and the sale price $P_r$. We can find from Figure 1 that with the increase of the retailer's implementation level of social responsibility, manufacturers' efforts to develop and improve product quality and the wholesale price of emergency medical supplies will increase. The $\beta$ will motivate the manufacturer to improve product quality and improve product supply efficiency. At the same time, when $\beta$ value is fixed, the effort level to improve product quality $e$ without government subsidies is lower than the corresponding value with the government subsidy scenario, indicating that government subsidies help the manufacturer implement improved product designs and fulfill social responsibility. The results of Figure 1 are the same as those in Corollary 3.

Figure 2 depicts the impact of the implementation level of social responsibility on the manufacturer's profit, retailer's profit, and supply chain profit.

Figure 2 demonstrates that as the implementation level of the retailer's social responsibility increases, profits of emergency medical member enterprises, supply chain system utilities, and consumer surplus will increase accordingly. Simultaneously, when the social responsibility implementation level $\beta$ value is fixed, the profits of the manufacturer, retailer, and supply chain systems without government subsidies are greater than the corresponding values with government subsidies, indicating that government subsidies help profits of supply chain members and system increase, so that emergency medical member enterprises have no worries about assuming social responsibilities. The results of Figure 2 are the same as those in Corollary 3.

*5.3. Coordination Effect Analysis of Wholesale Price–Cost Sharing Joint Contract*

To display the coordination effect of the wholesale price–cost sharing contract, the paper further analyzes the impact of the joint contract on emergency medical member enterprises of the emergency medical supplies supply chain, as shown in Figures 3–5.

Figure 3 reflects the impact of retailer's social responsibility implementation level $\beta$ and cost-sharing ratio $\omega$ on the improvement product design effort $e$. As the retailer's social responsibility implementation level $\beta$ and cost-sharing ratio $\omega$ increase, the manufacturer's effort of improved product design increases. This shows that when the proportion of the cost shared by the retailer is relatively high, it can encourage the retailer to boost the level

of social responsibility implementation, which in turn will increase the manufacturer's effort in improving product design.

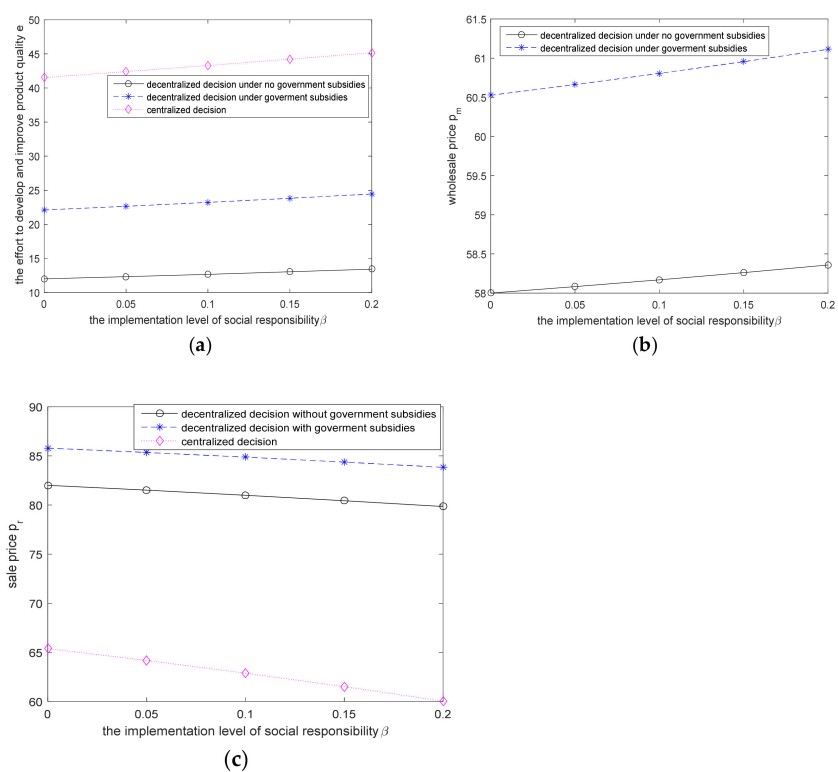

**Figure 1.** Impact $\beta$ on $e$, $P_m$, and $P_r$ (**a**) Impact of $\beta$ on $e$, (**b**) Impact of $\beta$ on $P_m$, (**c**) Impact of $\beta$ on $P_r$.

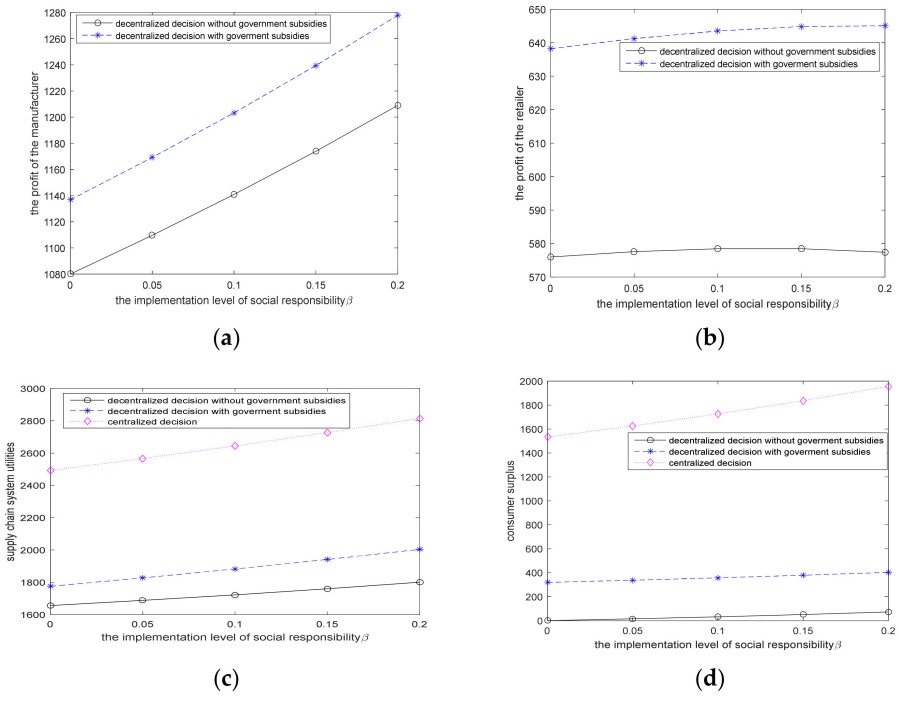

**Figure 2.** Impact of $\beta$ on $\pi_r$, $\pi_m$, $\pi_{sc}$, and $CS$ (**a**) Impact of $\beta$ on $\pi_m$, (**b**) Impact of $\beta$ on $\pi_r$, (**c**) Impact of $\beta$ on $\pi_{sc}$, (**d**) Impact of $\beta$ on $CS$.

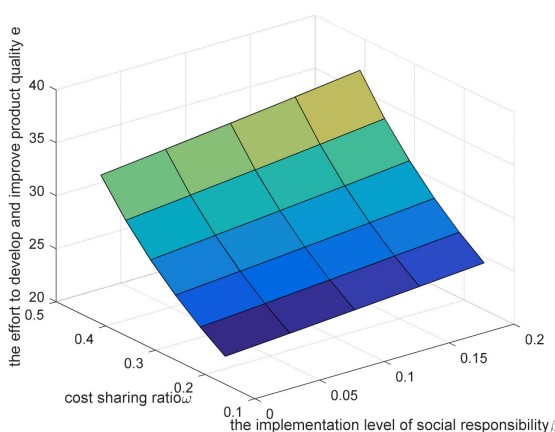

**Figure 3.** Impact of $\beta$ and $\omega$ on $e$.

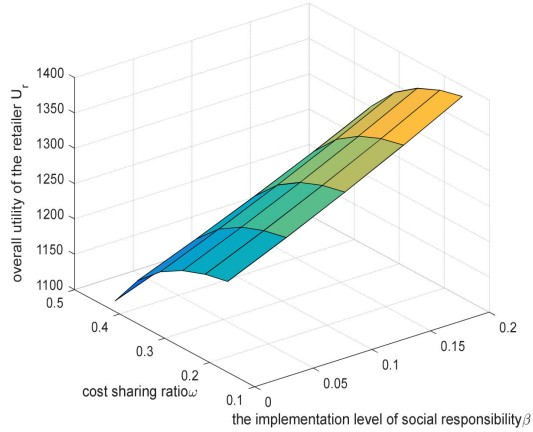

**Figure 4.** Impact of $\beta$ and $\omega$ on $U_r$.

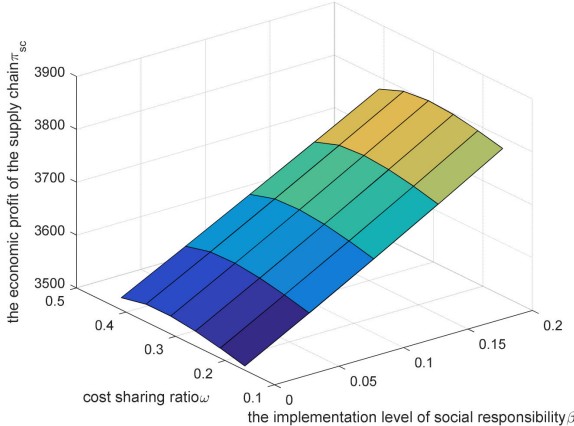

**Figure 5.** Impact of $\beta$ and $\omega$ on $\pi_{sc}$.

Figures 4 and 5 reflect the change in the overall profit of the emergency medical supply chain as the social responsibility implementation level $\beta$ and cost-sharing ratio $\omega$ change. As the fraction of cost-sharing increases, retailer's overall utility increases, and the economic profit of emergency medical supplies supply chain rose slowly at first and then declined slightly, showing an overall upward trend. As the retailer's social responsibility implementation level $\beta$ increases, the retailer's overall utility increases, and

the economic profit of the supply chain system also further increases. It indicates that the retailer can effectively share the manufacturer's cost and stimulate the retailer to perform social responsibilities more efficiently. Therefore, the retailer's overall utility and the economic profit of the supply chain system are improved.

Based on Figures 3–5, we can conclude that when the retailer shares the manufacturer's improved product design cost in a reasonable proportion, this can not only motivate itself to enhance the implementation level of social responsibility and their overall effectiveness but also increase the manufacturer's effort in improving product design and the economic profit of the supply chain system.

Finally, we will summarize the above results. Our findings confirm that the retailer's implementation level of social responsibility has a positive impact on the manufacturer and supply chain system. When the retailer's implementation level of social responsibility is in a certain range, that is, $\beta \in (0.33724, 1)$, it has a negative influence on the retailer. When $0 < \beta < 0.33724$, it has a positive impact on the retailer. Meanwhile, it has a negative influence on government subsidy and government subsidy directly affects manufacturers' technological innovation awareness. Therefore, the retailer's implementation level of social responsibility $\beta$ should be relatively small; utilities of emergency medical member enterprises and systems could be improved. Our insights confirm that the difficulty factor of manufacturer's technological innovation $\varepsilon$ negatively affects profits of emergency medical member enterprises and systems. To assume social responsibility and relieve the impact of the COVID-19 pandemic, the manufacturer should implement technological innovation. However, to maintain the balance, $\varepsilon$ should be relatively small. Our research also confirms that appropriate cost-sharing coefficient $\omega$ ($0.17 \leq \omega \leq 0.45$) can increase the optimal profit $\pi_m$ and the optimal utility $U_r$ in the joint contract decision scenario compared with the corresponding values in the decentralized decision scenario. In general, our research utilizes the mathematical model to extend the study of the emergency medical supply chain by taking into account the government subsidy, CSR, and social welfare.

## 6. Concluding Remarks with Future Scopes

### 6.1. Concluding Remarks

In view of the optimization of profit and social welfare, our paper comprehensively studies the factors, for example, game behavior, contract coordination, and the impact of the pandemic among enterprises in the emergency medical material supply chain and builds a three-echelon emergency medical supply chain consisting of the government, the manufacturer, and the retailer. The model explores the influence of factors such as changes in market demand due to the pandemic, the implementation level of the social responsibility, the effort to develop and improve product quality, and government subsidies on the optimal decision making of emergency medical member enterprises. Firstly, our paper analyzes the optimal decision making of members of the emergency medical supplies supply chain under the decentralized decision-making scenario in the case of no government subsidies. Secondly, our paper compares the optimal profit under different decision scenarios and introduces a joint contract to coordinate the emergency medical supplies supply chain. Finally, the comparative analysis of equilibrium results in the above-mentioned situations is carried out, and some conclusions can be drawn:

1.  Compared to equilibrium results without government subsidies, the manufacturer's effort to develop and improve product quality, the manufacturer's profit, the retailer's profit, and the total social welfare under the government subsidy situation are greater.
2.  The government subsidy coefficient to the manufacturer is not affected by the sensitivity of demand to manufacturer's effort to improve product design and supply efficiency λ, but λ has increased the effort of the manufacturer to develop and improve products, and the sale price of emergency medical supplies, thereby boosting the profits and social welfare of emergency medical member enterprises.
3.  Under the decentralized decision scenario, with the enhancement of implementation level of social responsibility, the government subsidy coefficient decreases, the manu-

facturer's effort to develop and improve product quality, the wholesale price, and the manufacturer's profit increases. The impacts of the retailer's implementation level of social responsibility on the sales price, retailer's profit, and the overall profit of the supply chain also depend on the manufacturer's effort to develop and improve product quality. Second, the government does not blindly subsidize the manufacturer, and its subsidy coefficient is directly related to the retailer's implementation level of social responsibility so that the total social welfare can be optimized.

4.  The technological innovation difficulty coefficient $\varepsilon$ directly affects the emergency medical supplies supply chain members and the optimal value of the system and the retailer's implementation of social responsibility. Therefore, to accomplish the coordination of the emergency medical supplies supply chain, the government can provide the subsidy to the manufacturer.

5.  When the fraction of cost-sharing is in a certain range, the profits of emergency medical member enterprises are higher than the corresponding values under decentralized decision making, and the joint contract can encourage emergency medical member enterprises to improve product design and fulfill social responsibilities, thereby boosting the profit of emergency medical member enterprises, realizing the reasonable allocation of supply chain profits.

*6.2. Managerial Implications*

Viewing the above analysis, our paper reveals some managerial implications:

1.  For the government: The government plays a vital role in the emergency medical supply chain during the COVID-19 pandemic situation. How to establish an effective incentive mechanism and encourage enterprises to assume social responsibility is a particularly important issue. From the perspective of the manufacturer, the government should provide different incentives, such as production or cost subsidies and various support policies, alleviating the uncertainty of manufacturing enterprises during the epidemic. Compared with no government subsidies, the utility of supply chain members under government subsidies has improved. Therefore, proper government subsidies not only help to maintain the balance of emergency medical supply chain enterprises but also to achieve unified management and save expenses. In terms of coordination models, government subsidy is positively correlated with the level of retailer's CSR implementation $\beta$, indicating that the government should focus on raising enterprises' CSR awareness.

2.  For the manufacturer: It is necessary to promote technological innovation level in terms of product quality, production efficiency, and material supply during the COVID-19 pandemic. However, technology investment will increase production costs and decrease the enterprise utility and the enthusiasm for social responsibility. In the process of fulfilling technological innovation to assume CSR, the manufacturer should pay attention to costs and the government needs to subsidize the manufacturer's technological innovation costs to reduce the burden and allow the manufacturer to assume more social responsibilities during the pandemic. Therefore, the manufacturer's technological innovation difficulty coefficient $\varepsilon$ should be relatively small to enhance awareness of social responsibility and obtain more profit.

3.  For the retailer: The retailer should enhance the CSR awareness and capability for cost-sharing $\omega$ for the manufacturer's technological innovation under the guidance of the government. Despite the higher CSR awareness and capability for cost-sharing $\omega$, the profit of the retailer is higher. However, the level of CSR implementation has a negative influence on government subsidies and directly affects the manufacturer's production investment decision. In addition, when the retailer has a range of cost-sharing capabilities ($0.17 \leq \omega \leq 0.45$) instead of a random range, the retailer will get more profits. Therefore, the retailer should not only pursue its economic interest but also undertake some CSR responsibilities by actively cooperating with its supply chain partners to maintain a certain level of supply chain CSR. Meanwhile, from Corollary

5, we can observe that centralized decision making is the best cooperation state. Therefore, emergency medical supply chain enterprises should balance technological innovation investments and CSR investments rather than blindly invest and strive to achieve the level of centralized decision making.

### 6.3. Limitations and Future Scopes

In summary, the coordination of the emergency medical supply chain in the pandemic can simultaneously consider goals of profit and social welfare, provide a certain decision reference for emergency medical member enterprises, and increase the fulfillment level of social responsibility in the emergency medical supply chain. However, the current work has several potential extensions. First of all, this study only uses the whole price–cost sharing joint contract for coordination and has not considered other types of contracts or the joint contract to coordinate the emergency medical supply chain under the pandemic. Furthermore, this research considers linear market demand, and subsequent research can be expanded to coordinate the emergency medical supply chain under non-linear market demand. Finally, this research only considers the manufacturer to undertake social responsibility through R&D investment. In the future, the manufacturer and the retailer may jointly invest and assume social responsibilities. This will be the direction of future research.

**Author Contributions:** Conceptualization, K.X.; methodology, S.Z.; resources, K.X.; writing—original draft preparation, S.Z.; writing—review and editing, P.G.; validation, S.Z. All authors have read and agreed to the published version of the manuscript.

**Funding:** This research was funded by the Open Project of Institute of Wuhan Studies, grant number IWHS20201002.

**Institutional Review Board Statement:** Not applicable.

**Informed Consent Statement:** Not applicable.

**Data Availability Statement:** Data is contained within the article. The data presented in this study are available in "A Game-Theoretic Approach for the CSR emergency medical supply chain during the COVID-19 crisis".

**Conflicts of Interest:** The authors declare no conflict of interest.

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
