# Peer review of "A Game-Theoretic Approach for CSR Emergency Medical Supply Chain during COVID-19 Crisis"

_sustainability, doi:10.3390/su14031315_

Round 1
Reviewer 1 Report
Dear Authors,
Your research is interesting and challenging. I have to congratulate for the theoretical methodological approach...It is well documented, written and with rigorous mathematical conclusions.
The literature research part is also well documented, although it seems to need more references from other parts of the world.
In first page you mention that ”economic growth rate has dropped by 1.5% ”. You probably mean by 1.5 pp.
In what it concerns the empirical research part your research still has a significant way to undergo. You do not mention any database that you use to verify your mathematical formulas (without the empirical research how can you prove that your mathematical findings check into the real economy?).
Also, you have to double-check your results using a set of listed companies, as the response of the stock exchange markets to any improvement solution is the most realistic one in the real world.
You mention that section 5 is dedicated to the calculation examples and parameter sensitivity analysis. However you do not mention any (relevant) database for any given company/companies, which reduces the value of your research. In lack of any other given objective landmark the financial markets are still the best factor for proving/disproving any given theoretical assumption. If you do not check the validity of your sound theoretical findings for a dataset from real economy it can remain an unvalidated theory.
These said I recommend improving your research starting with Section 5 to fully or partially validate your theory with a dataset from real economy (preferably listed companies). Otherwise it can be the object a relevant academic debate but not so much a validated research with data from real economy and financial markets.
In the fourth and fifth parts of your paper you have to mention how your findings relate to previous studies (they are in-line, different or they present a totally new approach).
After you perform such improvements I consider that your paper can be a serious candidate for publishing in this prestigious journal.
Best Regards
Author Response
Response to Reviewer 1 Comments
Thank you for the reviewer’s comments concerning our manuscript entitled “A game theoretic approach for production decisions and social welfare maximization in the CSR emergency medical supply chain with government subsidies during COVID-19 situation” (ID: sustainability-1545783). Those comments are all valuable and very helpful for revising and improving our paper, as well as the important guiding significance to our researches. We have studied comments carefully and have made corrections which we hope to meet with approval. Revised portions are marked in red in the paper and use the "Track Changes" function in
Microsoft Word. The main corrections in the paper and the responses to the reviewer’s comments are as follows:
Point 1: The literature research part is also well documented, although it seems to need more references from other parts of the world.
Response 1: Considering the reviewer’s suggestion, we have added about eight pieces of literature from other parts of the world into the instruction part involved in lines 34-127.
Point 2: In first page you mention that ”economic growth rate has dropped by 1.5% ”. You probably mean by 1.5 pp.
Response 2: Thank you for pointing this out, we have modified the data into 1.5 pp in line 39.
Point 3: In what it concerns the empirical research part your research still has a significant way to undergo. You do not mention any database that you use to verify your mathematical formulas (without the empirical research how can you prove that your mathematical findings check into the real economy?).
Also, you have to double-check your results using a set of listed companies, as the response of the stock exchange markets to any improvement solution is the most realistic one in the real world.
You mention that section 5 is dedicated to the calculation examples and parameter sensitivity analysis. However, you do not mention any (relevant) database for any given company/companies, which reduces the value of your research. In lack of any other given objective landmark the financial markets are still the best factor for proving/disproving any given theoretical assumption. If you do not check the validity of your sound theoretical findings for a dataset from real economy it can remain an unvalidated theory.
These said I recommend improving your research starting with Section 5 to fully or partially validate your theory with a dataset from real economy (preferably listed companies). Otherwise it can be the object a relevant academic debate but not so much a validated research with data from real economy and financial markets.
Response 3: It is really true as the reviewer suggested that we need fully or partially validate our theory with a dataset from the real economy. Therefore, I have checked the relevant literature data and the market average of the real product-related parameters, adjusted the data to make it in line with the actual situation, and added the source quotation of the data, and added managerial implications in the sixth part to make it more verifiable in lines 722-781 in 5.1. Numerical Analysis of Equilibrium Results.
To further display the above results and verify the validity of the joint contract model, our paper implements sensitivity analysis on some important parameters using numerical simulation. The parameters are summarized in the form of two datasets in Table 3. Because of the limited access to real market data, the first datasets are set in accordance with those in previous studies [41,66]. The second datasets applied in this study have been selected as a control group.
Table 3. Values of parameters under two datasets.
|
Parameters |
|
|
|
|
|
|
First example |
100 |
1 |
0.5 |
10 |
1 |
|
Second example |
10000 |
12 |
10 |
200 |
16 |
Table 4. Impact of retailer’s CSR level on equilibrium results.
|
|
Decision variables |
Profits and social welfare |
||||||||||
|
|
|
|
|
|
|
|
|
|
|
|
|
|
|
First example |
0.1 |
60.81 |
84.87 |
23.22 |
0.424 |
1203.3 |
643.51 |
679.26 |
1846.8 |
357.50 |
228.76 |
163.40 |
|
0.2 |
61.11 |
83.83 |
24.45 |
0.42 |
1277.8 |
645.07 |
725.71 |
1922.9 |
403.17 |
250.74 |
192.87 |
|
|
0.3 |
61.45 |
82.64 |
25.82 |
0.41 |
1362.0 |
641.27 |
778.68 |
2003.3 |
458.05 |
275.78 |
229.82 |
|
|
0.4 |
61.83 |
81.27 |
27.33 |
0.407 |
1457.8 |
629.70 |
839.60 |
2087.5 |
524.75 |
276.72 |
304.40 |
|
|
0.5 |
62.26 |
79.68 |
29.03 |
0.40 |
1567.7 |
606.87 |
910.30 |
2174.6 |
606.87 |
337.15 |
337.15 |
|
|
0.6 |
62.73 |
77.80 |
30.94 |
0.39 |
1695.1 |
567.56 |
993.22 |
2262.6 |
709.45 |
374.63 |
416.26 |
|
|
0.7 |
63.28 |
75.57 |
33.10 |
0.38 |
1844.1 |
503.83 |
1091.6 |
2348.0 |
839.71 |
417.37 |
521.71 |
|
|
0.8 |
63.89 |
72.87 |
35.55 |
0.37 |
2020.8 |
403.31 |
1210.0 |
2424.1 |
1008.3 |
465.64 |
665.19 |
|
|
0.9 |
64.59 |
69.55 |
38.35 |
0.35 |
2233.1 |
246.25 |
1354.4 |
2479.3 |
1231.3 |
518.97 |
864.94 |
|
|
Second example |
0.1 |
615.60 |
812.47 |
237.45 |
0.42 |
831210 |
516740 |
545450 |
1347900 |
287080 |
19135 |
683.40 |
|
0.3 |
628.69 |
805.21 |
268.86 |
0.41 |
958250 |
534160 |
648620 |
1492400 |
381540 |
23929 |
997.04 |
|
|
0.5 |
645.60 |
794.14 |
309.44 |
0.40 |
1128900 |
529500 |
794250 |
1658400 |
529500 |
30642 |
1532.1 |
|
|
0.7 |
668.16 |
776.20 |
363.59 |
0.38 |
1368500 |
466880 |
1011600 |
1835400 |
778140 |
40288 |
2518.0 |
|
|
0.9 |
699.38 |
744.77 |
438.50 |
0.35 |
1725100 |
247310 |
1360200 |
1972400 |
1236600 |
54292 |
4524.3 |
|
Table 5. Impact of manufacturer’s technological innovation difficulty coefficient on equilibrium results ( ).
|
|
Decision variables |
Profits and social welfare |
||||||||||
|
|
|
|
|
|
|
|
|
|
|
|
|
|
|
First example |
0.4 |
74.85 |
102.65 |
79.41 |
0.417 |
1667.6 |
1030 |
1201.7 |
2697.7 |
686.69 |
1051.0 |
840.83 |
|
0.6 |
66.54 |
90.77 |
46.15 |
0.417 |
1453.8 |
782.84 |
913.31 |
2236.7 |
521.89 |
532.54 |
426.04 |
|
|
0.8 |
63.13 |
85.90 |
32.53 |
0.417 |
1366.3 |
691.36 |
806.59 |
2057.6 |
460.91 |
352.74 |
282.19 |
|
|
1 |
61.28 |
83.26 |
25.12 |
0.417 |
1318.6 |
643.97 |
751.30 |
1962.6 |
429.31 |
262.85 |
210.28 |
|
|
1.2 |
60.11 |
81.59 |
20.45 |
0.417 |
1288.6 |
615.03 |
717.53 |
1903.7 |
410.02 |
209.19 |
167.36 |
|
|
1.4 |
59.31 |
80.45 |
17.25 |
0.417 |
1268.1 |
595.54 |
694.79 |
1863.6 |
397.03 |
173.63 |
138.90 |
|
|
1.6 |
58.73 |
79.61 |
14.92 |
0.417 |
1253.0 |
581.52 |
678.44 |
1834.6 |
387.68 |
148.35 |
118.68 |
|
|
Second example |
12 |
679.90 |
885.57 |
391.75 |
0.417 |
1042100 |
676800 |
789600 |
1718900 |
451200 |
38368 |
1534.7 |
|
14 |
646.98 |
838.55 |
312.76 |
0.417 |
970590 |
587150 |
685000 |
1557700 |
391430 |
28530 |
1141.2 |
|
|
16 |
625.11 |
807.31 |
260.27 |
0.417 |
923110 |
531100 |
619620 |
1454200 |
354070 |
22581 |
903.23 |
|
|
18 |
609.53 |
785.04 |
222.87 |
0.417 |
889270 |
492880 |
575020 |
1382100 |
328590 |
18627 |
745.09 |
|
|
20 |
597.86 |
768.38 |
194.87 |
0.417 |
863930 |
465190 |
542730 |
1329100 |
310130 |
15823 |
632.92 |
|
Besides, we also have added 6.2 Managerial Implications from lines 932 to 971 in the latest version as follows:
Based on the above analyses, our paper reveals some managerial implications:
(1) For the government: The government plays a vital role in the emergency medical supply chain during the COVID-19 pandemic situation. How to establish an effective incentive mechanism and encourage enterprises to assume social responsibility is a particularly important issue. From the perspective of the manufacturer, the government should provide different incentives, such as production or cost subsidies and various support policies, alleviating the uncertainty of manufacturing enterprises during the epidemic. compared with no government subsidies, the utility of supply chain members under government subsidies has improved. Therefore, proper government subsidies not only help to maintain the balance of emergency medical supply chain enterprises but also to achieve unified management and save expenses. In terms of coordination models, government subsidy is positively correlated with the level of retailer’s CSR implementation , indicating that the government should focus on raising enterprises’ CSR awareness.
(2) For the manufacturer. It is necessary to promote technological innovation level in terms of product quality, production efficiency, and material supply during the COVID-19 pandemic situation. However, because technology investment will increase production costs and decrease the enterprise utility and the enthusiasm for social responsibility. In the process of fulfilling technological innovation to assume CSR, the manufacturer should pay attention to cost and the government needs to subsidize the manufacturer’s technological innovation costs to reduce the burden and allow the manufacturer to assume more social responsibilities during the epidemic. Therefore, the manufacturer’s technological innovation difficulty coefficient ε should be relatively small to enhance awareness of social responsibility and obtain more profit.
(3) For the retailer: The retailer should enhance the CSR awareness and capability for cost-sharing for the manufacturer’s technological innovation under the guidance of the government. Although the higher the CSR awareness and capability for cost-sharing , the higher the profit of the retailer. However, the level of CSR implementation has a negative impact on government subsidies and directly affects the manufacturer’s production investment decision. In addition, when the retailer has a range of cost-sharing capabilities ( ) instead of a random range, the retailer will get more profits. Therefore, the retailer should not only pursue its economic interest but also should undertake some CSR responsibilities by actively cooperating with its supply chain partners, so as to maintain a certain level of supply chain CSR. Meanwhile, from Corollary 5, we can observe that centralized decision-making is the best cooperation state. Therefore, emergency medical supply chain enterprises should balance technological innovation investments and CSR investments rather than blindly invest, and strive to achieve the level of centralized decision-making.
Point 4: In the fourth and fifth parts of your paper you have to mention how your findings relate to previous studies (they are in-line, different or they present a totally new approach).
Response 4: As the reviewer suggested that the fourth and fifth parts need some relations(they are in-line, different or they present a totally new approach), we have added some contents from lines 708 to 721 at the end of the fourth parts:
Finally, we will compare our work with previous studies. Our findings confirm that when the government subsidizes manufacturer’s technological innovation effort, the profits of supply chain members and social welfare have increased. When the government’s subsidy for the manufacturer’s technological innovation effort reach a certain value, the utilities of supply chain members after contract coordination are higher than that of decentralized decision-making and reach the level of centralized decision-making [11]. We also extend this research by considering CSR. Our research also con-firms that appropriate effort of undertaking CSR can increase the profits of supply chain members and social welfare [13]. Besides, our work also confirms that revenue sharing-cost sharing can lead to perfect coordination [55]. However, our work discuss-es the manufacturer’s cost-sharing, not the retailer’s cost-sharing. In general, we utilize the game model to extend this research in view of government subsidy, CSR, social welfare, and COVID-19 and enrich emergency medical supply chain operation decisions under the pandemic.
Regarding the relation between Table or Figure and Corollary: we have added the relations as follows and the relations are reflected in the corresponding position of the paper as follows:
corollary 3 is supported by the results of Table 3. in line 748
The results of Table 4 are the same as those in Corollary 2. in line 768
The results of Table 5 are the same as those in equation (62). in line 780
The results of Figure 1 are the same as those in Corollary 3. in line 813
The results of Figure 2 are the same as those in Corollary 3. in line 829
We appreciate for reviewer’s warm work earnestly and hope that the correction will meet with approval. Once again, thank you very much for your comments and suggestions.

Reviewer 2 Report
Dear Authors.
Thanks for submitting your paper to the MDPI journal Sustainability.
Your paper concerns the medical supply chain management during the covid 19 crisis.
This article is pleasant reading, and I particularly enjoyed doing the review. However, I must state that as it is, it is not enjoyable and appreciable by a broader audience that may be not familiar with game-theoretic studies. Therefore, although I believe this study will be an interesting publication, it requires some minor checks and adjustments to be ready for a broader audience.
TITLE
I believe the title is excessively long. I understand that the paper involves many variables, but the title creates confusion. Try to focus on the essential part of the paper. A title like:
“A Game-Theoretic Approach for emergency medical supply chain during the covid-19 crisis”. Would it work for you?
ABSTRACT
Line 18-24, I believe the sentences need a minor rephrasing. “we can obtain” does just not sound good. I would opt for some other construct. Probably starting the sentence with “we observe (1)….” It would sound better.
INTRODUCTION
The introduction is almost complete; just it strikes the eyes that from lines 52 to 130, there are no citations. It is necessary to insert specific citations when new data is presented along with anecdotes or news events. Please also add a citation at the end of the first sentence at line 36.
Line 93 and line 115 it seems that the same idea is repeated. Is it necessary?
Since there are many variables to consider and many questions to answer, it would be helpful for the reader to have a scheme at the end of the introduction that summarizes:
-What are the literature considered in this paper
-What are the research questions
-What is the methodology
-What are the expected/desired outcomes of the study.
Otherwise, it is quite hard to follow the study.
LITERATURE REVIEW
The literature review is quite exhaustive; however, in the end, it would be interesting to specify a bit more what are the contribution of this study in relation to other papers. The authors already provide those but at a general level. A more specific description of the enhancement with respect to specific prior studies would be helpful to contextualize this paper with relation to previous works.
MODEL
The model description is quite clear and intuitive; however, it lacks a bit of relation with past literature. Could you please add a clear link with prior literature if similarities in the model construction are used?
I mean, when developing the dynamic game, are the choices or parameters unlinked to prior works? if so, please provide citations for that. Otherwise, the study seems totally unlinked to prior works, and it sounds weird.
RESULTS
The result description is clear and exhaustive from an economic point of view. However, as it often happens, it lacks a bit more effort from the authors to make results more available for a practitioner audience. I suggest the authors then add at the end of results paragraphs a detailed scheme that summarizes
The question that is being answered
The related mathematic (e.g., (2), (3), Corollary 2)
The intuition behind the mathematics
CONCLUSIONS
Would you please expand the section related to contribution for practitioners? How they can benefit from the results of this paper.
FINAL THOUGHTS
This paper is a good work of literature. With some minor efforts, it could be ready to be published.
Good luck with your research!
Author Response
Response to Reviewer 2 Comments
Thank you for the reviewer’s comments concerning our manuscript entitled “A game theoretic approach for production decisions and social welfare maximization in the CSR emergency medical supply chain with government subsidies during COVID-19 situation” (ID: sustainability-1545783). Those comments are all valuable and very helpful for revising and improving our paper, as well as the important guiding significance to our researches. We have studied comments carefully and have made corrections which we hope to meet with approval. Revised portions are marked in red in the paper and use the "Track Changes" function in
Microsoft Word. The main corrections in the paper and the responses to the reviewer’s comments are as follows:
Point 1: TITLE
I believe the title is excessively long. I understand that the paper involves many variables, but the title creates confusion. Try to focus on the essential part of the paper. A title like:
“A Game-Theoretic Approach for emergency medical supply chain during the covid-19 crisis”. Would it work for you?
Response 1: According to the Reviewer’s suggestion and paper content, we have changed the title to “A Game-Theoretic Approach for the CSR emergency medical supply chain during the covid-19 crisis”.
Point 2: ABSTRACT
Line 18-24, I believe the sentences need a minor rephrasing. “We can obtain” does just not sound good. I would opt for some other construct. Probably starting the sentence with “we observe (1) ….” It would sound better.
Response 2: According to the Reviewer’s suggestion, we have replaced “we can obtain” with “we observe” in the abstract.
Point 3:INTRODUCTION
The introduction is almost complete; just it strikes the eyes that from lines 52 to 130, there are no citations. It is necessary to insert specific citations when new data is presented along with anecdotes or news events. Please also add a citation at the end of the first sentence at line 36.
Line 93 and line 115 it seems that the same idea is repeated. Is it necessary?
Since there are many variables to consider and many questions to answer, it would be helpful for the reader to have a scheme at the end of the introduction that summarizes:
-What are the literature considered in this paper
-What are the research questions
-What is the methodology
-What are the expected/desired outcomes of the study.
Otherwise, it is quite hard to follow the study.
Response 3: Thank you for pointing this out, we have inserted specific citations from lines 35 to 127 and we have deleted the same idea from lines 116 to 118.
Regarding the third point (There need have a scheme at the end of the introduction), we have made modifications in accordance with the above requirements from Lines 148 to 179 in the latest version:
More specifically, our research adopts Shu et al. [14] and Li et al. [15] work on supply chain coordination from the perspective of the government and enterprises assuming social responsibility. they argue that CSR and government subsidy have a positive effect on supply chain decisions and the goal of maximizing social welfare can help increase the profits of supply chain companies. Accordingly, integrating government subsidy and enterprises’ CSR awareness into the emergency medical supply chain and considering their impacts on operational decisions of the emergency medical supply chain are meaningful theoretically and empirically. Most emergency supply chain coordination considers economic benefits, whereas social welfare has rarely been quantitatively examined. To bridge this gap, we model an emergency medical supply chain engaged in CSR and government subsidy composed of one manufacturer and one retailer. The manufacturer fulfills CSR through technological innovation effort. The retailer fulfills CSR through consumer surplus and may share part of the manufacturer’s technological innovation effort cost. Meanwhile, the government improves the determination of the manufacturer in technological innovation to fulfill CSR through cost subsidy to the manufacturer. As a result, four emergency medical models including decentralized decision models without and with government subsidies, centralized decision model, and wholesale price-cost sharing joint contract decision models are established. In the following sections, we explore the impacts of CSR implementation level and technological innovation effort on the utilities of emergency medical member enterprises and systems and analyze the relationships among governments, enterprises, and society, so as to provide insights for government and corporate decision-making and have emergency management capabilities against emergencies in the pandemic. Specifically, some points are proposed and answered.
(1) What are the effects of the retailer’s implementation level of social responsibility on profits of emergency medical member enterprises and systems?
(2) Whether the retailer’s implementation level of social responsibility will affect government subsidy and consumer surplus?
(3) What are the effects of the difficulty factor of manufacturer’s technological innovation ε on profits of emergency medical member enterprises and systems?
(4) Whether the decentralized supply chain can be coordinated and how it is coordinated?
Point 4: LITERATURE REVIEW
The literature review is quite exhaustive; however, in the end, it would be interesting to specify a bit more what are the contribution of this study in relation to other papers. The authors already provide those but at a general level. A more specific description of the enhancement with respect to specific prior studies would be helpful to contextualize this paper with relation to previous works.
Response 4: Considering the Reviewer’s suggestion, we have added more specific explanations to illustrate the contribution of this study and enhance the connection between this paper and previous works involved in lines 231-234, lines 270-276, lines 311-314, and lines 333-361 in the latest version. The specific additions are as follows:
Lines 231-234: The discussion above shows that the previous works on uncertainty in the supply chain include dual-channel supply chain, reverse supply chain, closed-loop supply chain, and other supply chains, which are the result of emergency incidents. Scholars have seldom considered decision-making in the emergency medical supply chain.
Lines 270-276: Reviewing the above literature, we can find out that the CSR of the enterprise is reflected in the improvement of technical level and emission capacity. But few scholars consider enterprise’s effort to improve product quality into market demand function. Meanwhile, a growing number of studies in the emergency medical supply chain inte-grates social welfare maximization (SWM), considering CSR, which can be the future research orientation. Meanwhile, the paper explores the impacts of CSR on the optimal strategies, enterprises utilities, and social welfare.
Lines 311-314: It is evident that with a growing body of literature on emergency supply chains, some scholars consider the uncertainty of demand or supply, while others integrate government intervention and consumer surplus. This paper points out that the government subsidies the cost of enterprise efforts.
Lines 333-361: Table 1 summarizes the previous literature related to this paper. To sum up, the above literature has conducted an in-depth analysis of emergency supply chain coordination, social welfare, CSR, and contract coordination, and achieved absolute results. However, the above literature has never simultaneously considered government policies, social welfare, CSR, emergency medical supply chain management, and contract coordination. This paper fills the gap by comprehensively integrating these factors into the coordination model of the emergency medical supply chain. Besides, most of the literature analyses emergency supply chain coordination issues, only considering economic and environmental responsibilities. Due to the influence of the pandemic, emergency medical supply chain coordination not only considers economic factors but also focuses on corporate social responsibilities and social welfare.
Table 1. Comparison between this study and prior research.
|
Reference |
SC structure |
CSR |
Demand influence factor |
Game approach |
Coordination mechanism |
Governments Policies |
Social Welfare |
|
Chen and Su(2019) |
1M+1PA |
|
price |
MS-led PA-led |
RS contract |
√ |
√ |
|
Zhou et al. (2018) |
1M+nR |
|
Price, carbon emissions, and substitutability degree |
M-led |
|
√ |
√ |
|
Bai et al. (2021) |
1M+1R |
√ |
Price, CSR level, and emission technology level |
R-led |
RCS contract |
|
|
|
Wang et al. (2021) |
1M+1R |
√ |
price |
M-led |
‘GSCS contract |
√ |
|
|
Liu et al. (2021) |
1M+1R |
√ |
Price, CSR investment level, and competition coefficient between two retailers |
M-led |
RCS contract |
|
|
|
Cheng et al. (2021) |
1M+1R |
√ |
Price, corporate reputation |
M-led |
|
|
|
|
Shu et al. (2018) |
1M+1R |
√ |
price |
M-led |
|
|
√ |
|
Asl-Najaf et al. (2021) |
1M+1R |
|
Price and product amount |
|
TT contract |
|
|
|
Hong et al. (2016) |
1M+1Re |
|
|
M-led |
|
|
|
|
Mondal et al. (2021) |
|
|
Random, COVID-19 |
|
|
|
|
|
Jiang et al. (2021) |
1PP+nF |
|
Random |
|
WP and QP contract |
√ |
|
|
Reza Rezayat et al. (2021) |
2M+2R |
|
Price and quality |
M-led |
|
|
√ |
|
Feng et al. (2021) |
OEM+IR |
|
Random |
|
|
√ |
|
|
Current study |
1M+1R |
√ |
Price, CSR investment level |
M-led |
WPCS contract |
√ |
√ |
Note: √=covered; S=supplier; M=manufacturer; R=retailer; Re=recycler; PA=Photovoltaic system assembler; MS= module supplier; PP= power plant; F=farmer; OEM=original equipment manufacturer; IR=independent remanufacturer; RS=revenue sharing; RCS=revenue and cost sharing; GSCS=government subsidy and cost sharing; TT=Two-part tariff; WP= wholesale price; QP=quantity payment; WPCS=wholesale price and cost sharing.
Point 5: MODEL
The model description is quite clear and intuitive; however, it lacks a bit of relation with past literature. Could you please add a clear link with prior literature if similarities in the model construction are used?
I mean, when developing the dynamic game, are the choices or parameters unlinked to prior works? if so, please provide citations for that. Otherwise, the study seems totally unlinked to prior works, and it sounds weird.
Response 5: It is really true as the reviewer suggested that a clear link between the model and the previous is missing here, we have added the serial number of the corresponding reference to the model from lines 380 to 418 in the latest version.
Point 6: RESULTS
The result description is clear and exhaustive from an economic point of view. However, as it often happens, it lacks a bit more effort from the authors to make results more available for a practitioner audience. I suggest the authors then add at the end of results paragraphs a detailed scheme that summarizes
The question that is being answered
The related mathematic (e.g., (2), (3), Corollary 2)
The intuition behind the mathematics
Response 6: according to the Reviewer’s comments, we have added the detailed scheme that summarizes the question that is being answered, the related mathematic, and the intuition behind the mathematics from lines 867 to 885 at the end of section 5 as follows:
Regarding the first and third points:
Finally, we will summarize the above results. Our findings confirm that the retailer’s implementation level of social responsibility has a positive impact on the manufacturer and supply chain system. When the retailer’s implementation level of social responsibility is in a certain range, that is, , it has a negative impact on the retailer. When , it has a positive impact on the retailer. Meanwhile, it has a negative impact on government subsidy and government subsidy directly affects the manufacturer’s technological innovation awareness. Therefore, the retailer’s implementation level of social responsibility should be relatively small, utilities of emergency medical member enterprises and systems could be improved. Our insights confirm the difficulty factor of manufacturer’s technological innovation negatively affects profits of emergency medical member enterprises and systems. To assume social responsibility and relieve the impact of the COVID-19 pandemic, the manufacturer should implement technological innovation. However, to maintain the balance, should be relatively small. Our research also confirms that appropriate cost-sharing coefficient can increase the optimal profit and the optimal utility in the joint contract decision scenario compared with the corresponding values in the decentralized decision scenario. In general, our research utilizes mathematical model to extend the study of the emergency medical supply chain by considering government subsidy, CSR and social welfare.
Regarding the second point: we have added the relations between Table or Figure and Corollary as follows and the relations are reflected in the corresponding position of the paper as follows:
corollary 3 is supported by the results of Table 3. in line 748
The results of Table 4 are the same as those in Corollary 2. in line 768
The results of Table 5 are the same as those in equation (62). in line 780
The results of Figure 1 are the same as those in Corollary 3. in line 813
The results of Figure 2 are the same as those in Corollary 3. in line 829
Point 7: CONCLUSIONS
Would you please expand the section related to contribution for practitioners? How they can benefit from the results of this paper.
Response 7: As Reviewer suggested that conclusion section requires more detailed contribution for practitioners. We have added 6.2 Managerial Implications from lines 932 to 971 in the latest version as follows:
Based on the above analyses, our paper reveals some managerial implications:
(1) For the government: The government plays a vital role in the emergency medical supply chain during the COVID-19 pandemic situation. How to establish an effective incentive mechanism and encourage enterprises to assume social responsibility is a particularly important issue. From the perspective of the manufacturer, the government should provide different incentives, such as production or cost subsidies and various support policies, alleviating the uncertainty of manufacturing enterprises during the epidemic. compared with no government subsidies, the utility of supply chain members under government subsidies has improved. Therefore, proper government subsidies not only help to maintain the balance of emergency medical supply chain enterprises but also to achieve unified management and save expenses. In terms of coordination models, government subsidy is positively correlated with the level of retailer’s CSR implementation , indicating that the government should focus on raising enterprises’ CSR awareness.
(2) For the manufacturer. It is necessary to promote technological innovation level in terms of product quality, production efficiency, and material supply during the COVID-19 pandemic situation. However, because technology investment will increase production costs and decrease the enterprise utility and the enthusiasm for social responsibility. In the process of fulfilling technological innovation to assume CSR, the manufacturer should pay attention to cost and the government needs to subsidize the manufacturer’s technological innovation costs to reduce the burden and allow the manufacturer to assume more social responsibilities during the epidemic. Therefore, the manufacturer’s technological innovation difficulty coefficient ε should be relatively small to enhance awareness of social responsibility and obtain more profit.
(3) For the retailer: The retailer should enhance the CSR awareness and capability for cost-sharing for the manufacturer’s technological innovation under the guidance of the government. Although the higher the CSR awareness and capability for cost-sharing , the higher the profit of the retailer. However, the level of CSR implementation has a negative impact on government subsidies and directly affects the manufacturer’s production investment decision. In addition, when the retailer has a range of cost-sharing capabilities ( ) instead of a random range, the retailer will get more profits. Therefore, the retailer should not only pursue its economic interest but also should undertake some CSR responsibilities by actively cooperating with its supply chain partners, so as to maintain a certain level of supply chain CSR. Meanwhile, from Corollary 5, we can observe that centralized decision-making is the best cooperation state. Therefore, emergency medical supply chain enterprises should balance technological innovation investments and CSR investments rather than blindly invest, and strive to achieve the level of centralized decision-making.
We appreciate for reviewer’s warm work earnestly and hope that the correction will meet with approval. Once again, thank you very much for your comments and suggestions.

Reviewer 3 Report
Dear authors
Thank you for the work.
The abstract of the current work is ambiguous and its sentences are long and very difficult to understand. Research questions are not clearly formulated in the introduction. In addition, the flows of the ideas and text throughout the paper are not smooth due to the heavy usages of long sentences and the chaotic structure of the paper. Presentation strategy should be substantially improved.
The mathematical writing of the model is poor. For example, equation (1) has two terms (Q and a) with the same definition, i.e. Market demand for emergency medical supplies. The parameters are defined twice, one time in the table 1 and explained again under equation (1). this makes the paper very lengthy due to several duplications.
It is not also clear what is the type of the developed model. There is also no clear connection between sections, for example section (3) and section (4). In section 3, equation (1) is described without stating what is its purpose and why it is defined in section 3 that according to its title should describe (Model Descriptions and Assumptions).
The presentation, writing and description of the developed model are very poor. The model is very hard to understand.
Although the paper studies the social responsibility, this paper focuses more on the mathematics and is within the scope of the Operation Research (OR) studies. So, I think that it is not a good fit to the sustainability journal.
Author Response
Response to Reviewer 3 Comments
Thank you for the reviewer’s comments concerning our manuscript entitled “A game theoretic approach for production decisions and social welfare maximization in the CSR emergency medical supply chain with government subsidies during COVID-19 situation” (ID: sustainability-1545783). Those comments are all valuable and very helpful for revising and improving our paper, as well as the important guiding significance to our researches. We have studied comments carefully and have made corrections which we hope to meet with approval. Revised portions are marked in red in the paper and use the "Track Changes" function in
Microsoft Word. The main corrections in the paper and the responses to the reviewer’s comments are as follows:
Point 1:
The abstract of the current work is ambiguous and its sentences are long and very difficult to understand.
Response 1: According to the Reviewer’s suggestion and paper content, we have modified some long sentences of the abstract into short sentences for easy understanding as follows in lines 8-30.
Abstract: The pandemic has caused high fluctuations in the demand to medical supplies. Hence, emergency medical supplies enterprises have faced challenges in the decision-making and need to consider more corporate social responsibility (CSR) in production. At the same time, the government needs to take some measures to support emergency medical supplies enterprises. Motivated by this issue, our paper investigates the decision and coordination problems for emergency medical supply chain considering CSR with the government, a manufacturer, and a retailer. The manufacturer produces emergency medical supplies and has more production technological innovation effort to improve supply efficiency and assume CSR; the retailer faces uncertain demand and is responsible for undertaking CSR to meet demand; the government needs to implement a certain degree of subsidies to alleviate the impact of the pandemic on emergency medical supply chain enterprises. Meanwhile, our paper further explores the three responsibilities of the economy, society, and efficiency of enterprises under the COVID-19 pandemic and the decision-making of enterprises to implement CSR. Based on the principle of maximizing social welfare, we discuss decentralized decision-making (without government and with the government) and centralized decision-making, respectively. On this basis, our paper designs a wholesale price-cost sharing joint contract coordination mechanism and proves that the joint contract can achieve supply chain coordination under certain conditions. Through the analysis, we observe: (1) Government subsidies can improve the enthusiasm of supply chain members to undertake CSR; (2) With the improvement of the retailer's CSR level, the profits of supply chain members and overall performance have improved to varying degrees; (3) To improve supply efficiency and assume social responsibility, the manufacturer implement technological innovation investment, but it will impose a certain burden on the manufacturer. Government subsidies can allow the manufacturer to achieve a balance between social responsibility and its profit.
Point 2: Research questions are not clearly formulated in the introduction.
Response 2: Considering the Reviewer’s suggestion, we have added several descriptions, including the literature considered in this paper, research questions, the methodology, and the expected outcomes of the study at the end of the introduction in Lines 148-179 as follows:
More specifically, our research adopts Shu et al. [14] and Li et al. [15] work on supply chain coordination from the perspective of the government and enterprises assuming social responsibility. they argue that CSR and government subsidy have a positive effect on supply chain decisions and the goal of maximizing social welfare can help increase the profits of supply chain companies. Accordingly, integrating government subsidy and enterprises’ CSR awareness into the emergency medical supply chain and considering their impacts on operational decisions of the emergency medical supply chain are meaningful theoretically and empirically. Most emergency supply chain coordination considers economic benefits, whereas social welfare has rarely been quantitatively examined. To bridge this gap, we model an emergency medical supply chain engaged in CSR and government subsidy composed of one manufacturer and one retailer. The manufacturer fulfills CSR through technological innovation efforts. The retailer fulfills CSR through consumer surplus and may share part of the manufacturer’s technological innovation effort cost. Meanwhile, the government improves the determination of the manufacturer in technological innovation to fulfill CSR through cost subsidy to the manufacturer. As a result, four emergency medical models including decentralized decision models without and with government subsidies, centralized decision model, and wholesale price-cost sharing joint contract decision models are established. In the following sections, we explore the impacts of CSR implementation level and technological innovation effort on the utilities of emergency medical member enterprises and systems and analyze the relationships among governments, enterprises, and society, so as to provide insights for government and corporate decision-making and have emergency management capabilities against emergencies in the pandemic. Specifically, some points are proposed and answered.
(1) What are the effects of the retailer’s implementation level of social responsibility on profits of emergency medical member enterprises and systems?
(2) Whether the retailer’s implementation level of social responsibility will affect government subsidy and consumer surplus?
(3) What are the effects of the difficulty factor of manufacturer’s technological innovation ε on profits of emergency medical member enterprises and systems?
(4) Whether the decentralized supply chain can be coordinated and how it is coordinated?
Point 3: the flows of the ideas and text throughout the paper are not smooth due to the heavy usages of long sentences and the chaotic structure of the paper. Presentation strategy should be substantially improved.
Response 3: Thank you for pointing this out, we have asked people whose working language is English to help revise the paper to make the paper smoother.
Point 4: The mathematical writing of the model is poor. For example, equation (1) has two terms (Q and a) with the same definition, i.e. Market demand for emergency medical supplies. The parameters are defined twice, one time in table 1 and explained again under equation (1). this makes the paper very lengthy due to several duplications.
It is not also clear what is the type of the developed model. There is also no clear connection between sections, for example section (3) and section (4). In section 3, equation (1) is described without stating what is its purpose and why it is defined in section 3 that according to its title should describe (Model Descriptions and Assumptions).
Response 4: It is really true as the reviewer suggested that the relevant definitions of section 3 are not clearly described and there needs a clear link between sections, we have rewritten model descriptions and assumptions in section 3 from lines 362 to 418. Besides, we have added some clarification at the beginning of section 4 to reinforce the connection between section 3 and section 4 from lines 420 to 427.
from lines 362 to 418
Model Descriptions and Assumptions
Our paper establishes an emergency medical supplies supply chain system consisting of a manufacturer and a retailer considering the maximization of profits and social welfare. Sudden increase and decrease in the demand for emergency medical supplies due to the pandemic require the manufacturer and the retailer to perform social responsibilities to increase the production of emergency medical supplies. The manufacturer produces emergency medical supplies and has more production technological innovation effort to improve supply efficiency and assume CSR; the retailer faces uncertain demand and is responsible for undertaking CSR to meet demand; to encourage supply chain members to invest in product improvement to enhance supply efficiency and assume CSR, the government also gives manufacturers certain special subsidy.
The dynamic game sequence is: (1) The government intends to maximize social welfare and gives the manufacturer a certain special subsidy; (2) The dominant manufacturer provides emergency medical supplies, determines the R&D investment of improved products, and determines its wholesale price ; (3) The retailer in the subordinate position purchases emergency medical supplies from the manufacturer, determines the implementation level of corporate social responsibility (CSR), and sells them at a certain retailer price .
Assumption 1. referencing the literature [38,41,57]. the demand is a function of the retailer price and the manufacturer’s investment efforts in the emergency medical supply chain, which can be represented as
(1)
Where and constantï¼› . stands for the base market size, refers to the sensitivity coefficient of demand to the sale price of emergency medical supplies, refers to the sensitivity coefficient of demand to manufacturer’s effort to improve product design and improve supply efficiency during the pandemic, and refers to the manufacturer’s effort to enhance product design to improve the efficiency of emergency medical supplies.
Assumption 2. In addition to basic production cost, the manufacturer is working hard on research and development, adopting alternative designs to increase delivery speed to fulfill CSR. Generally speaking, the research and development cost for improved products is nonlinearly increasing over . Assume that the basic production cost is and the additional R&D investment cost is denoted as , which is a quadratic cost function. represents the difficulty coefficient of technological innovation, The greater the , the more difficult it is to develop alternative designs, and the greater the R&D investment required. This cost function is similar to that in literature [41,58,59]. As the regulator of emergency handling, the government’s intervention can affect the supply chain members’ decisions and it sets a proportion to share the member’s effort cost [15,64,65]. Let represent the government subsidy rate and the government directly subsidizes the manufacturer according to its investment effort, denoted as .
Assumption 3. Learning from the implementation of CSR in the references [60,61], This section discusses how retailers implement CSR and utilize consumer surplus to express the CSR effect. At this time, the retailer aims to maximize the utility , , where is the retailer’s profit and is the retailer’s concern degree for consumer surplus, that is, is the level of CSR implementation. In addition, consumer surplus refers to the difference between the highest price consumers are willing to pay for the product and the actual market price paid, which is also commonly used in the literature [14,40,62]. Therefore, consumer surplus ( ) is:
(2)
Social welfare is part of the objective function of the supply chain. To effectively encourage the manufacturer to resume work and production, the government intends to maximize social welfare and subsidizes effort cost to develop and improve the product quality of the manufacturer. Similar to the literature [37,63], According to economic assumptions, social welfare consists of four parts: manufacturer’s profit , retailer’s profit , consumer surplus , and total government subsidy expenditure . Therefore, the social welfare function is expressed: , , that is
(3)
from lines 420 to 427
- Decision Model of Emergency Medical Supply Chain During the Pandemic
In this section, based on the Stackelberg game, we explore the optimal decisions under the decentralized scenario without or with government subsidy and centralized scenario. For convenience, the superscript , , represent the case without government subsidy under the decentralized scenario, the case with government subsidy under the decentralized scenario, and the case with government subsidy under the centralized scenario, respectively. Then, based on wholesale price-cost sharing joint contract and government subsidy, we will construct a coordination mechanism considering CSR and government subsidy in the next section.
We appreciate for reviewer’s warm work earnestly and hope that the correction will meet with approval. Once again, thank you very much for your comments and suggestions.

Round 2
Reviewer 1 Report
Dear Authors,
You have responded to most of my suggestions and recommendations. Still the one with checking your assumptions for real data from the market is not yet addressed.
I guess the paper can be a good candidate for publishing now although a proper academic research needs empirical confirmation (otherwise we can just imagine that our theory is correct, without any real life proof...).
Best regards